

**Secondary Organic Aerosols from OH Oxidation of Cyclic Volatile Methyl Siloxanes as an Important**
**Si Source in the Atmosphere**
Chong Han[1,2], Hongxing Yang[2], Kun Li[1,3], Patrick Lee[1], John Liggio[1], Amy Leithead[1], Shao-Meng Li[4]*
[1]Air Quality Research Division, Environment and Climate Change Canada, Toronto, Ontario M3H 5T4,
Canada
[2]School of Metallurgy, Northeastern University, Shenyang, 110819, China
[3]Laboratory of Atmospheric Chemistry, Paul Scherrer Institute, Villigen 5232, Switzerland
[4]State Key Joint Laboratory of Environmental Simulation and Pollution Control, College of Environmental
Sciences and Engineering, Peking University, Beijing, China 100871
**Correspondence**: Shao-Meng Li (shaomeng.li@pku.edu.cn)
**Short summary:** We presented yields and compositions of Si-containing SOA generated from the reaction
of cVMS (D3-D6) with OH radicals. $NO_x$ played negative roles on cVMS SOA formation, while ammonium
sulfate seeds enhanced D3-D5 SOA yields at short photochemical ages under high-$NO_x$ conditions. The
aerosol mass spectra confirmed that the components of cVMS SOA significantly relied on OH exposure. A
global cVMS-derived SOA source strength was estimated to understand SOA formation potentials of cVMS.
**Abstract**
Cyclic volatile methyl siloxanes (cVMS) are active ingredients in widely used consumer products, which
can volatilize into the atmosphere, thus attracting much attention due to their potential environmental risks.
While in the atmosphere the cVMS undergo oxidation yielding both gaseous and particulate products. The
aerosol yields and compositions from the OH oxidation of four cVMS (D3-D6) were determined under low



and high-NO$_x$ conditions in an oxidation flow reactor. The aerosol yields progressively increased from D3
to D6, consistent with the volatilities and molecule weights of these cVMS. NO$_x$ can restrict the formation
of SOA, leading to lower SOA yields under high-NO$_x$ conditions than under low-NO$_x$ conditions, with a
yield decrease between 0.05-0.30 depending on the cVMS. Ammonium sulfate seeds exhibited minor
impacts on SOA yields under low-NO$_x$ conditions, but significantly increased the SOA yields in the oxidation
of D3-D5 at short photochemical ages under high-NO$_x$ conditions. The mass spectra of the SOA showed a
dependence of its chemical compositions on OH exposure. At high exposures, equivalent to photochemical
ages of >6 days in the atmosphere, D4-D6 SOA mainly consisted of C$_x$H$_y$ and C$_x$H$_y$O$_z$Si$_n$ under low-NO$_x$
conditions, whereas they primarily contained N$_m$O$_z$, C$_x$H$_y$, C$_x$H$_y$O$_1$, C$_x$H$_y$O$_{>1}$ and C$_x$H$_y$O$_z$Si$_n$ under high-NO$_x$
conditions. Using the yield data from the present study and reported cVMS annual production, a global
cVMS-derived SOA source strength is estimated to be 0.16 Tg yr$^{-1}$, distributed over major urban centers.

**1 Introduction**
Secondary organic aerosols (SOA), which contribute 50-85% to the mass of atmospheric organic aerosols
(OA) (Glasius and Goldstein, 2016), are mainly formed via the partitioning of low volatility products from
oxidation of volatile organic compounds (VOCs), semi- and intermediate volatile organic
compounds(S/IVOCs) (Riipinen et al., 2012). SOA has attracted significant attention due to their important
impacts on climate, ecosystems and human health (Berndt et al., 2016). Global budgets of SOA remain an
unresolved issue despite extensive research, largely due to uncertainties associated with aerosol yields and
the presence of unconsidered SOA precursors.
As one type of anthropogenic VOC and potential SOA precursors, cyclic volatile methyl siloxanes (cVMS)
are widely used in industrial applications and personal care products (Genualdi et al., 2011; Krogseth et al.,



2013a). cVMS have been classified as high-volume chemicals with an annual production of millions of tons
globally (Rücker and Kümmerer, 2015; Ahrens et al., 2014). Studies of cVMS in the environment have
focused on investigating health and environmental impacts particularly due to their potential persistence,
bioaccumulation and toxicity (Guo et al., 2019; Liu et al., 2018; Farasani and Darbre, 2017; Xu et al., 2019;
Kim et al., 2018; Coggon et al., 2018). As a result, the European Council has proposed a restriction on the
octamethylcyclotetrasiloxane (D4) and decamethylcyclopentasiloxane (D5) content in wash-off personal
care products to a limit of 0.1 mass% by 2020. The legislative actions notwithstanding, knowledge of
environmental behavior of cVMS remains surprisingly scarce as compared to their applications and
economic significance (Rücker and Kümmerer, 2015).

54       It has been estimated that approximately 90% of cVMS are emitted into the atmosphere due to their high

saturation vapor pressures (Allen et al., 1997). Gas-phase cVMS have been observed in both indoor and
outdoor air. Tang et al. (2015) reported that cVMS accounted for about one third of total VOC mass
concentration in a classroom. Outdoor air concentrations of cVMS have also been measured at different sites
worldwide (Li et al., 2020; Wang et al., 2018; Rauert et al., 2018), increasing from rural to urban sites and
consistent with increasing population density (Rücker and Kümmerer, 2015). For example, at a rural site in
Sweden, the concentration of hexamethylcyclotrisiloxane (D3), D4, D5 and dodecamethylcyclohexasiloxane
(D6) were 0.94, 3.5, 13 and 1 ng/m$^3$, respectively (Kierkegaard and Mclachlan, 2013), while they were 18,
55, 172 and 14 ng/m$^3$ in urban areas of Toronto in Canada, respectively (Genualdi et al., 2011; Rauert et al.,
2018). cVMS have also been detected in the remote Arctic atmosphere, confirming their long-range transport
(Genualdi et al., 2011; Krogseth et al., 2013b). Atmospheric half-lives of cVMS are on the order of 30, 15,
10 and 1.6 days for D3-D6, respectively, which allow cVMS to exhibit a hemispherical distribution in the
atmosphere (Whelan et al., 2004; Allen et al., 1997; Yucuis et al., 2013; Xiao et al., 2015; Macleod et al.,



2013). These lifetimes are driven mostly by reactions with the OH radicals (Xiao et al., 2015; Wang et al.,
2013), which generate silanols and dimeric products that can be partitioned to condensed phases (Coggon et
al., 2018; Sommerlade, 1993; Wu and Johnston, 2016). The loss of cVMS in the atmosphere is slight through
$O_3$ and $NO_3$ due to their small reaction rates, and by Cl atoms on account of its low concentration (Atkinson,
1991; Alton and Browne, 2020).
It has been demonstrated that elemental Si is a frequent constituent of nanoparticles in rural and urban
areas (Phares et al., 2003; Rhoads, 2003; Bein, 2005; Bzdek et al., 2014) and in remote regions (Li and
Winchester, 1990; Li and Winchester, 1993). These Si-containing nanoparticles have previously been
attributed to ore smelting processes, but recent studies have shown that Si-containing species are one of the
main components in cVMS SOA, suggesting that the oxidation of cVMS may be an important source of Si
in atmospheric aerosols (Wu and Johnston, 2016, 2017). In a modeling study, the oxidation products of
cVMS (D4, D5 and D6) were considered to quantify the maximum potential for aerosol formation through
reactions with the OH radicals (Janechek et al., 2017). Chandramouli and Kamens (2001) demonstrated the
gas-particle partitioning of silanols from D5 oxidation by the OH radicals. Wu and Johnston (2016, 2017)
analyzed the chemical composition of secondary aerosols from OH oxidation of D4 and D5, showing a large
number of monomeric and dimeric products. Janechek et al. (2017, 2019) reported physical properties of
SOA generated by OH oxidation of D5, including hygroscopicity, cloud seeding potential and volatility.
Charan et al. (2021) measured SOA yields of D5 using chambers and flow tube reactors, indicating necessary
conditions when extrapolating SOA yields to the ambient atmosphere. These studies mainly focused on D5
and occasionally D4 but rarely others. To better understand the SOA-forming potentials of typical cVMS in
the atmosphere, accurate yields and molecular compositions of SOA from the oxidation of cVMS under
various atmospheric conditions are needed.



In this work, the formation of SOA from the oxidation of four cVMS (D3-D6) by OH radical was
investigated in an oxidation flow reactor (OFR). Under various combinations of $NO_x$ and ammonium sulfate
seed concentrations, the yields and compositions of SOA formed from the oxidation were measured using a
suite of instruments including a scanning mobility particle sizer (SMPS), a proton transfer reaction time of
flight mass spectrometer (PTR-ToF-MS) and an aerosol mass spectrometer (AMS). Based on these SOA
yields, the contribution of cVMS to SOA in the global atmosphere was estimated using reported cVMS
concentrations. The results obtained here can largely improve our understanding of the contribution and
composition of SOA from cVMS.
**2 Experiments and methods**
**2.1 Photo-oxidation experiments**
The reactions of cVMS with OH radicals were controlled at a constant temperature ($21\pm1\,^{\circ}$C) and relative
humidity ($35\%\pm2\%$) in a custom-made oxidation flow reactor (the Environment and Climate Change Canada
oxidation flow reactor, ECCC-OFR), which is shown in Fig. S1 of the Supplement and has been described
in detail previously (Li et al., 2019a). Briefly, the ECCC-OFR is a fused quartz cylinder (length: 50.8 cm,
inner diameter: 20.3 cm) equipped with a conical inlet and 7 outlets. Wall losses of particles and gases in the
ECCC-OFR have been shown to be lower than in other OFRs (Huang et al., 2017; Lambe et al., 2011;
Simonen et al., 2017; Li et al., 2019a). The length and full angle of cone inlet are 35.6 cm and 30°,
respectively, designed to minimize the formation of jetting and recirculation in the OFR. The outlet at the
reactor center is a stainless-steel sampling port (inner diameter: 0.18 in) extending 12.7 cm long into the
ECCC-OFR. This sampling inlet reduces the impact of potential turbulent eddies caused by the back end of
the reactor. The remaining 6 outlets around the perimeter are designed to allow side flows to pass through
the OFR as a sheath flow, indirectly reducing wall losses of gases and particles inside the OFR upon sampling.





Ozone-free mercury UV lamps for generating OH radicals are housed in small quartz tubes around and in
parallel to the quartz reaction cylinder, and a large flow of air through each of these smaller quartz tubes is
used to remove the heat produced by the lamps. The relative humidity was adjusted by controlling the ratio
of dry air to wet air into the reactor, and was measured using a humidity sensor (Vaisala) at one of the sheath
flow outlets (side flows) of the reactor. The volume of the entire ECCC-OFR is about 16 L and the total flow
rate is 8 L min$^{-1}$, leading to a residence time of 2 min in the OFR.
OH radicals were produced through the reaction of water vapor with O($^1$D) formed from O$_3$ photolysis at
254 nm. The OH concentration in the ECCC-OFR was regulated by controlling the input voltage and the
number of UV lamps. Methanol vapor, introduced into the ECCC-OFR through a bubbler containing
methanol solution, was used to determine the OH exposure (i.e., photochemical age) by tracking its decay
in the reactor from the reaction with the OH. The decay, or fractional loss, of gas-phase methanol,
[MeOH]/[MeOH]$_0$ was measured with the PTR-ToF-MS, and was used to calculate the OH concentration
via Equation 1,
$$[OH]=-\frac{1}{k_{MeOH}}\ln\frac{[MeOH]}{[MeOH]_0} \tag{1}$$

where $k_{MeOH}$ is the second-order rate constant of methanol reaction with OH at 298 K (9.4×10$^{-13}$ cm$^3$
molecule$^{-1}$ s$^{-1}$). It was noted that the OH exposure measurement was offline, because methanol can affect the
OH reactivity with cVMS. Under low and high-NO$_x$ conditions described below, the OH exposure varied in
the range of 5.5 × 10$^{10}$-1.8 × 10$^{12}$ and 8.2 × 10$^{10}$-1.1 × 10$^{12}$ molecules cm$^{-3}$ s, respectively. They correspond
to 0.4-14.2 and 0.6-8.5 equivalent days (photochemical age), respectively, assuming that an average OH
concentration in air is 1.5 × 10$^6$ molecules cm$^{-3}$ (Mao et al., 2009).
Pure D3-D6 cVMS compounds (solid D3 and liquid D4, D5 and D6) were placed in a glass U-type tube
and maintained at the room temperature. Vapors from these compounds (Table S1 of the Supplement) were





separately introduced into the ECCC-OFR by a small flow of zero air (1-18 mL min$^{-1}$) passing over the
headspace of the U-tube containing the pure compounds. The concentrations of D3-D6 in the ECCC-OFR
ranged from 20 to 40 ppb, depending on their volatilities. To study the influence of existing particles on the
SOA formation, ammonium sulfate (AS) seed particles were produced using an atomizer, dried by a diffusion
dryer and neutralized by a neutralizer and injected into the reactor without size selection. The mass
concentration of AS seed particles was approximately 30 μg m$^{-3}$ for all experiments.
$N_2O$ was used as a source of NO to achieve high-$NO_x$ conditions (Lambe et al., 2017). $NO_x$ conditions
were defined by the fate of peroxy radicals ($RO_2$), which was described by the reaction rate ratio ($R_{NO}$) of
$RO_2$ + NO and $RO_2$ + $HO_2$ (Peng et al., 2017). The $R_{NO}$ ratio increases with increasing OH exposures at a
constant concentration of $N_2O$ (Li et al., 2019b). To achieve a constant branching ratio during the high-$NO_x$
experiments, the initial $N_2O$ concentration in the OFR was varied (1.6%-8.0%) to maintain an $R_{NO}$ value of
20 (Li et al., 2019b), as calculated using a model (OFR Exposures Estimator v3.1) (Peng et al., 2017). A
ratio of $R_{NO}$=20 indicates that 95% of $RO_2$ reacts with NO, ensuring the dominance of $RO_2$ + NO, which
represents conditions that are relevant for urban atmosphere (Peng et al., 2019). Under low-$NO_x$ conditions,
$N_2O$ was not introduced into the OFR, where the $RO_2$ radicals dominantly interacted with $HO_2$ radicals,
representing atmospheric scenarios with few $NO_x$ sources.
**2.2 Characterization and analysis**
The concentrations of cVMS in the OFR were measured online with a proton transfer reaction time of
flight mass spectrometer (PTR-ToF-MS, Ionicon Analytik GmbH) (Liggio et al., 2016). The number and
mass size distribution of aerosols was monitored using a scanning mobility particle sizer (SMPS, TSI). The
mass spectra and elemental composition of aerosols was determined with a high-resolution time-of-flight
aerosol mass spectrometer (HR-ToF-AMS, Aerodyne) and analyzed with the AMS analysis software Squirrel



(version 1.62G) and Pika (version 1.22G).
SOA mass yields ($Y$) were calculated via Equation 2,
$$Y = \frac{\Delta C_{\text{SOA}}}{\Delta C_{\text{cVMS}}} \qquad (2)$$
where $\Delta C_{\text{SOA}}$ and $\Delta C_{\text{cVMS}}$ are the mass concentrations of SOA formed and cVMS lost, respectively. The mass
concentration of SOA was determined by multiplying the effective aerosol density by the integrated SOA
volume concentration from the SMPS, subtracting the AS seed volume for experiments with AS seeds. The
effective aerosol density ($\rho$) was calculated for unseeded experiments through the following Equation 3
(Lambe et al., 2015),
$$\rho = \frac{D_{va}}{D_m} \qquad (3)$$
where $D_{va}$ is the vacuum aerodynamic diameter obtained from the HR-ToF-AMS, and $D_m$ is the electric
mobility diameter measured by the SMPS. The $\rho$ varied in the range of 1.6-1.8 depending on the cVMS. The
same $\rho$ value was also used in the seeded experiments.
**3 Results and discussion**
**3.1 SOA yields**
Figure 1 shows the SOA yields from the photooxidation of the D3-D6 cVMS under low and high-NO$_x$
conditions as a function of photochemical age (PA), i.e., time-integrated OH exposure, with and without AS
seed particles. SOA yields have been widely used to estimate the potential of precursors to produce aerosol
mass (Mcfiggans et al., 2019; Li et al., 2019a; Bruns et al., 2015; Lambe et al., 2015). As shown in Fig. 1,
the cVMS SOA yields exhibited an overall increasing trend with PA, expressed in equivalent photochemical
days, which agreed with the trend of D5 SOA yields reported by Janechek et al. (2019). Under low-NO$_x$
conditions (Fig. 1a), SOA yields exhibited a slow growth, reaching a plateau after 10 equivalent days. This
may be due to increased gas-phase fragmentation of cVMS to generate some higher volatility products,





leading to a small increasing amplitude of partition ratio of species into SOA at longer photochemical ages.

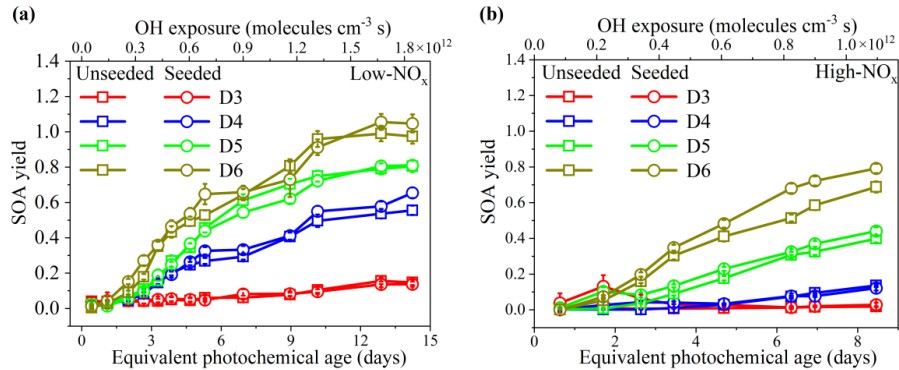

Figure 1. SOA yields from unseeded and seeded (30 μg m$^{-3}$) photooxidation of cVMS by OH radicals. **(a)**
low-NO$_x$ experiments; **(b)** high-NO$_x$ experiments.

For the unseeded and low-NO$_x$ experiments in Fig. 1a, SOA yields of four cVMS exhibited significant
differences in values over the same number of equivalent days. The SOA yields successively increased from
D3 to D6, consistent with the volatilities and molecular masses of the cVMS as well as their reaction rate
constants with the OH radical (Alton and Browne, 2020; Kim and Xu, 2017; Safron et al., 2015). The
maximum SOA yields of D3-D6 were (0.16±0.02), (0.56±0.03), (0.80±0.03) and (0.99±0.04), respectively,
occurring after a PA of 12 equivalent days. It has been reported that D5 SOA yields varied in the range of
0.08-0.50 (Janechek et al., 2019; Wu and Johnston, 2017). Under a similar OH exposure (1.6-1.7×10$^{12}$
molecules cm$^{-3}$ s), the D5 SOA yield (0.79±0.03) obtained here is considerably higher than that (0.22)
measured by Janechek et al. (2019), which may be attributed to differences in experimental conditions such
as differences in wall losses, SOA measurement methods, RHs, and D5 concentrations (Table S2). Although
the amount of cVMS lost was variable, cVMS SOA yields positively depended on SOA mass concentrations
(Fig. S2), and this trend was observed in previous D5 SOA experiments with OH oxidation (Wu and Johnston,

194     2017).





As shown in Fig. 1b, the order of SOA yields from the four cVMS under high-$NO_x$ conditions was the
same as that under low-$NO_x$ conditions. However, the SOA yields under high-$NO_x$ conditions were generally
smaller than the corresponding yields at similar OH exposures under low-$NO_x$ conditions, with a decrease
of 0.05-0.30 depending on the cVMS. Such a reduction suggests that $NO_x$ can restrict the formation of cVMS
SOA. $NO_x$ has been shown to reduce SOA yields for some anthropogenic alkanes (Li et al., 2019b), aromatics
(Ng et al., 2007a; Chan et al., 2009; Zhou et al., 2019), monoterpenes (Zhao et al., 2018) and other terpenes
(Ng et al., 2007b), attributable to the formation of higher volatility products (e.g., organic nitrates) generated
by $RO_2$ + NO compared to $RO_2$ + $HO_2$ (Presto et al., 2005; Li et al., 2019b), which is also likely the case
here. The higher volatility products favor partitioning in the gas phase, thus reducing the potential for
forming SOA (Zhou et al., 2019). Moreover, high $NO_x$ levels can suppress the formation of products for
nucleation, thereby reducing aerosol surface as a condensational sink and increasing the wall loss of
condensable species in an OFR under high-$NO_x$ conditions (Zhao et al., 2018; Sarrafzadeh et al., 2016; Wildt
et al., 2014). Figure S3 indicates that the difference between SOA yields with and without $NO_x$ decreased
with increasing silicon atoms within individual cVMS, indicating a less restricting effect of $NO_x$ on the SOA
formation for larger cVMS. This means that high $NO_x$ levels play a lessor role in the SOA yields of lower
volatility precursors.
SOA yields in the AS-seeded experiments under low and high-$NO_x$ conditions are also shown in Figs. 1
and S3, indicating minimal impacts of the AS seed particles on SOA yields. A yield enhancement ratio
($R_E$=$Y_{seeded}$/$Y_{unseeded}$, Fig. S4) was used to show the seed impacts more clearly. Under low-$NO_x$ conditions,
the $R_E$ values for all cVMS were close to 1.0 (Fig. S4a), suggesting negligible impact of AS seed particles
on SOA yields; however, under high-$NO_x$ conditions, $R_E$ was much larger (17.81, 13.18 and 15.51 for D3-
D5, respectively) at short PA but gradually decreased to 1.0 with increasing PA for D3-D5, while it was





always close to 1.0 for D6 regardless of PA (Fig. S4b). $R_E$ values greater than 1.0 suggest that AS seed
particles played an enhancement role in the cVMS SOA formation, as similarly reported in SOA formation
from hydrocarbons (Sarrafzadeh et al., 2016; Lamkaddam et al., 2017; Li et al., 2019b). Under low-$NO_x$
conditions, the general lack of impact on cVMS SOA yields by the AS seed particles suggests that
condensation was not the main process driving SOA formation in cVMS oxidation. For the few cases of high
$NO_x$ level at low PA, where $R_E$ was >1 for D3-D5, it is possible that their early generations of oxidation
products were more volatile than successive generations of products and hence more prone to condensation
enhanced by AS seeds. As PA increased, further reactions of these early generation oxidation products with
OH radicals resulted in further generation products that were likely less volatile, thereby weakening the
enhancing role of AS seeds at high OH exposure. Such effect was less pronounced for D6, likely because its
oxidation products at different PA had similar volatilities. Figure S4b shows that the effect of AS seed
particles on SOA yields negatively correlated with the number of silicon atoms in the cVMS. Lower volatility
precursors (D5 and D6) formed lower volatile products (Alton and Browne, 2020), resulting in SOA yields
less sensitive to the pre-existing seeds. In fact, the oxidation products of D5 have been shown to be nearly
non-volatile (Janechek et al., 2019; Wu and Johnston, 2017).
**3.2 Aerosol compositions**
**3.2.1 Compositions of SOA under low-$NO_x$ conditions.**
Figures 2 and S5 show the normalized HR-ToF-AMS mass spectra of cVMS SOA from unseeded
experiments under low-$NO_x$ conditions at OH exposures of $9.0 \times 10^{11}$ molecules $cm^{-3}$ s. The mass spectral
signals can be identified as fragments with a formula of $C_xH_yO_zSi_n$. For D3 SOA, the most prominent peaks
were at m/z 44 and 29, dominated by $CO_2^+$ and $CHO^+$, which were tracers for organic acids (Ng et al., 2010),
alcohols and aldehydes (Lee et al., 2012), respectively. They may result from the oxidation of the methyl





groups in D3 by OH radicals. For the mass spectra of D4-D6 SOA, the two highest peaks at m/z 14 and 15
were $CH_2^+$ and $CH_3^+$, respectively. In addition, there were several dominant $C_xH_yO_zSi_n$ peaks, which were
fragments of silicon-containing products. For the $C_xH_yO_zSi_n$ group in D4 SOA, there were four typical peaks
at m/z 255, 257, 271 and 273, with formulae of $C_4H_{11}O_7Si_3^+$, $C_3H_9O_8Si_3^+$, $C_3H_7O_9Si_3^+$ and $C_3H_9O_9Si_3^+$,
respectively. The $C_xH_yO_zSi_n$ fragment group containing Si of D5 SOA had three dominant peaks at m/z 327,
329 and 331, corresponding to $C_{12}H_{11}O_2Si_5^+$, $C_9H_9O_8Si_3^+$ and $C_5H_{15}O_9Si_4^+$, respectively. For the $C_xH_yO_zSi_n$
group containing Si in D6 SOA, there were five main peaks at m/z 73 ($C_3H_9Si$), 387 ($C_8H_{23}O_8Si_5^+$), 389
($C_8H_9O_9Si_5^+$), 401 ($C_9H_{21}O_{10}Si_4^+$) and 403 ($C_7H_{15}O_{12}Si_4^+$).

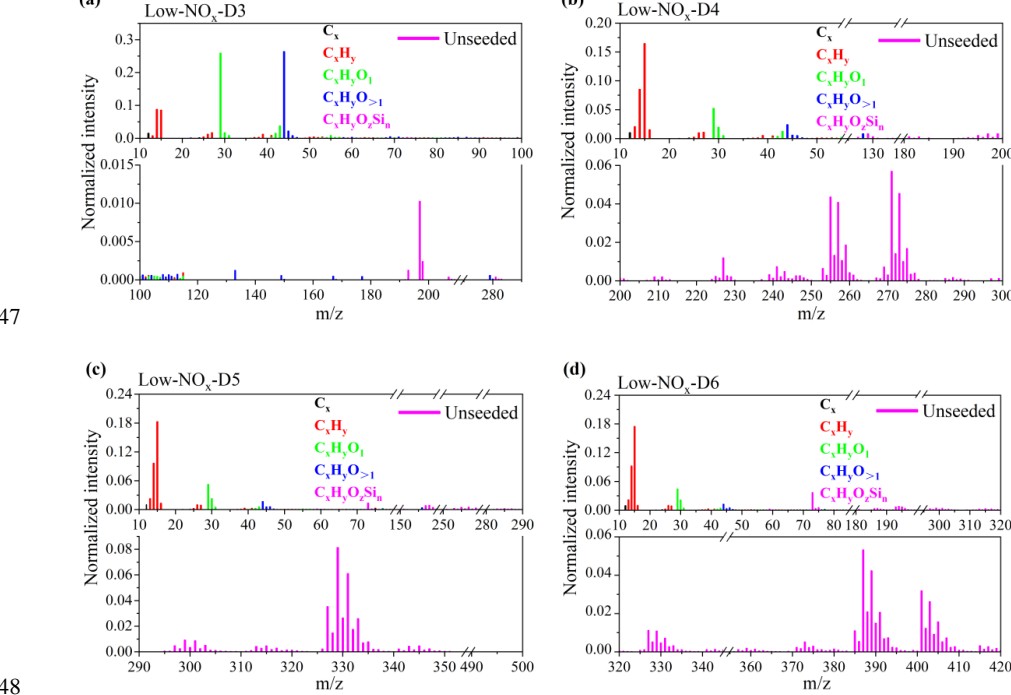



Figure 2. HR-ToF-AMS mass spectra of cVMS SOA at OH exposure of $9.0 \times 10^{11}$ molecules cm$^{-3}$ s under
low-NO$_x$ conditions in unseeded experiments. **a-d** represent the mass spectra of D3-D6 SOA, respectively.



Figure 3 shows the evolution of different groups of ions in the HR-ToF-AMS spectra of the cVMS SOA
as a function of PA in equivalent days. For D3-D6 under unseeded conditions, $C_xH_yO_1$ and $C_xH_yO_{>1}$ ions
significantly decreased within 0-4 equivalent days of PA, but remained essentially unchanged when PA
increased to 7-15 equivalent days. The $C_xH_y$ ion increased to its peak value at about 9 equivalent days of PA
for D3 and 2-3 equivalent days of PA for D4-D6, and then gradually decreased with further PA increases.
The $C_xH_yO_zSi_n$ group of ions maintained an increasing trend until 9-10 days of PA, thereafter it decreased
slightly for D4-D6 SOA. The weighted values of the atomic number ratios Si/C (n/x) and Si/O (n/z) for the
$C_xH_yO_zSi_n$ groups in D5 and D6 SOA at different PA are plotted in Fig. S6, which can be used to indicate
the changes in the Si element of SOA. The n/x ratio at initial SOA formation stage was close to that (0.50)
in D5 and D6 molecules, and then increased continuously with increasing PA. The n/z ratio kept increasing
from 0.53 to 1.15 for D5 SOA and from 0.72 to 1.32 for D6 SOA, but varying around 1.0 that was the Si/O
ratio in D5 and D6. While it is difficult to separate the effect of fragmentation due to the AMS ionization
process, the relative changes of group intensities and the evolution of n/x and n/z in $C_xH_yO_zSi_n$ over different
PA may be attributed to the initial substitution of methyl groups on the -Si-O- ring of the cVMS by OH or
$CH_2OH$ when oxidized by the OH radicals (Wu and Johnston, 2016; Alton and Browne, 2020). Si-containing
products such as silanols may partition into SOA and result in an appearance of $C_xH_yO_zSi_n$ ions in the AMS
mass spectra. It is possible that in this process, the Si-O bonds may continue to be cleaved from OH radical
attack, followed by further deposition of less volatile and nonvolatile products containing $C_xH_yO_zSi_n$ on the
SOA.



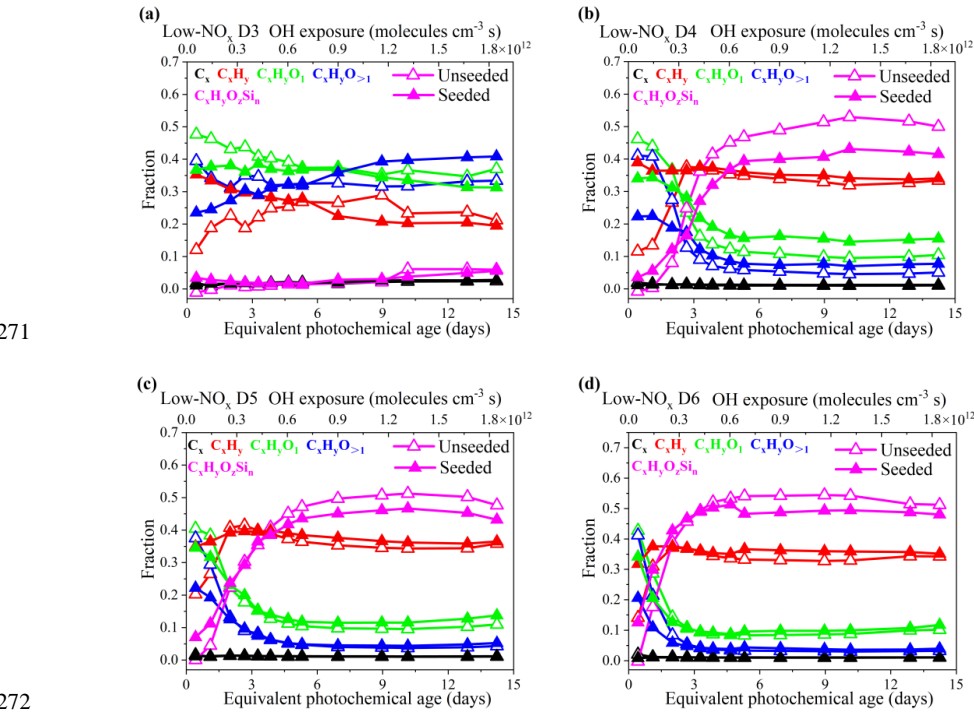



Figure 3. Fraction of $C_x$, $C_xH_y$, $C_xH_yO_1$, $C_xH_yO_{>1}$ and $C_xH_yO_zSi_n$ ion groups for SOA derived from the

oxidation of cVMS (**a-d**) by OH radicals at different photochemical ages under low-$NO_x$ conditions.

Empty and solid triangles represent experimental data under unseeded and seeded conditions, respectively.

As shown in Fig. 3, the presence of seeds led to some changes in the evolution trends of ion groups in the

AMS spectra. For instance, the initial fraction of $C_xH_y$ in seeded experiments was larger than that in unseeded

experiments, whereas $C_xH_yO_1$ and $C_xH_yO_{>1}$ exhibited opposite changes. The presence of seeds led to larger

initial and smaller steady-state $C_xH_yO_zSi_n$ fractions than those in unseeded experiments. Regardless of the

presence of seeds, $C_xH_y$, $C_xH_yO_1$ and $C_xH_yO_{>1}$ mainly contributed to the composition of all cVMS SOA at

initial OH radicals oxidation, but D4-D6 SOA primarily consisted of $C_xH_y$ and $C_xH_yO_zSi_n$ after 6 equivalent

days.



### 3.2.2 Compositions of SOA under high-NO$_x$ conditions.

Figure S7 shows the HR-ToF-AMS mass spectra of cVMS SOA under high-NO$_x$ conditions at OH

exposures of $9.0 \times 10^{11}$ molecules cm$^{-3}$ s (-6.9 d). Compared to that under low-NO$_x$ conditions (Figs. 2 and

S5), there was one additional N-containing group (N$_m$O$_z$) in the SOA mass spectra under high-NO$_x$

conditions, which accounted for 16%-31%. For the mass spectra of D3-D6 SOA originating from unseeded

experiments under high-NO$_x$ conditions in Fig. S7, the dominating peaks of the N$_m$O$_z$ family were m/z 30

(NO$^+$) and m/z 46 (NO$_2^+$). The common main peaks were located at m/z 30 (NO$^+$) for cVMS SOA, m/z 44

(CO$_2^+$) for D3-D4 SOA, and m/z 46 (NO$_2^+$) for D4-D6 SOA. In addition, there were other primary peaks at

m/z 29 (CHO$^+$) for D4 SOA, while m/z 15 (CH$_3^+$) and m/z 28 (CO$^+$) for D5-D6 SOA. The m/z 28 (CO$^+$),

similar with m/z 44 (CO$_2^+$), is considered as a tracer for organic acids (Ng et al., 2010). In the mass spectra

for D3-D6 SOA under high-NO$_x$ conditions, the presence of NO$^+$ (m/z 30) and NO$_2^+$ (m/z 46) illustrated the

formation of nitrates in SOA (Ng et al., 2007b).

For the C$_x$H$_y$O$_z$Si$_n$ group in the D4 SOA mass spectrum under high-NO$_x$ conditions, the dominating peaks

and their formulas were same as those under low-NO$_x$ conditions. For the C$_x$H$_y$O$_z$Si$_n$ group in D5 SOA, in

addition to two typical peaks at m/z 327 and 329 in the low-NO$_x$ experiments, there was another prominent

peak at m/z 328, with a formula C$_8$H$_{12}$O$_5$Si$_5$. The C$_x$H$_y$O$_z$Si$_n$ group in D6 SOA had three typical peaks at m/z

73 (C$_3$H$_9$Si), m/z 387 (C$_8$H$_{23}$O$_8$Si$_5^+$) and m/z 401 (C$_9$H$_{21}$O$_{10}$Si$_4^+$). For the C$_x$H$_y$O$_z$Si$_n$ groups in cVMS SOA,

there was little difference in the x, y, z and n value assignment of C$_x$H$_y$O$_z$Si$_n$ peaks in SOA generated under

low-NO$_x$ and high-NO$_x$ conditions, suggesting the formation of similar Si-containing oxidation products.

For cVMS SOA under high-NO$_x$ conditions, the evolution of family groups as a function of OH exposure

was summarized in Fig. S8. The dominated composition at initial stage was C$_x$H$_y$O$_{>1}$ groups for D3-D6 SOA.

At equivalent days larger than 6, D3 SOA primarily consisted of C$_x$H$_y$O$_{>1}$, N$_m$O$_z$ and C$_x$H$_y$O$_1$ groups, while



D4-D6 SOA was mainly composed of $C_xH_yO_zSi_n$, $N_mO_z$, $C_xH_yO_1$, $C_xH_y$ and $C_xH_yO_{>1}$ groups. Figure S8 also
shows influences of seeds on the evolution of family groups under high-$NO_x$ conditions. It was observed
that all groups in D3-D6 SOA displayed similar change trends regardless of the existence of seeds. As shown
in Fig. S9, the trend of the weighted values of the atomic ratio n/z in the $C_xH_yO_zSi_n$ groups at different
photochemical ages under high-$NO_x$ conditions was similar to that in low-$NO_x$ experiments. However, the
n/x ratios remained almost unchanged, and were close to the initial value (0.5) in cVMS. This may be
attributed to possible suppression of cleavage of methyl groups from the -Si-O- ring of the cVMS under
high-$NO_x$ conditions.
**4 Conclusions and implications**
The yields and compositions of SOA generated from the photooxidation of four cVMS (D3-D6) with OH
radicals were investigated using an oxidation flow reactor. cVMS SOA yields exhibited an overall increasing
trend with PA, and their values gradually increased with cVMS from D3 to D6. SOA formations depended
on $NO_x$, as shown by smaller SOA yields under high-$NO_x$ conditions. Ammonium sulfate seeds significantly
enhanced SOA yields of D3-D5 at short PA under high-$NO_x$ conditions. The SOA mass spectra showed that
Si-containing species were one of main chemical compositions at PA of >6 days.
To evaluate the potential contributions of cVMS to SOA formation in the atmosphere, global SOA
concentrations produced from cVMS were estimated according to the cVMS SOA yields measured in this
work and using the cVMS concentrations reported from multiple studies, which were listed in Table S3.
Here, under the seeded conditions, the high-$NO_x$ SOA yields at 8.5 equivalent days (D3: 0.028; D4: 0.122;
D5: 0.441; D6: 0.792) are employed in the calculation of cVMS SOA concentrations at urban sites, while
the low-$NO_x$ SOA yields under the unseeded conditions at 14.2 equivalent days (D3: 0.148; D4: 0.556; D5:
0.805; D6: 0.975) are used to estimate cVMS SOA at background and polar sites. Figure 4 shows the global





concentration distribution of SOA from four cVMS (D3-D6) at 36 sites worldwide estimated by the Equation

329    4,

$$C_{\text{cVMS-SOA}} = C_{\text{cVMS}} \times \frac{\Delta C_{\text{cVMS}}}{C_{\text{in-cVMS}}} \times Y \qquad (4)$$
where $C_{\text{cVMS}}$ and $C_{\text{cVMS-SOA}}$ are the mass concentration of cVMS reported from literatures and cVMS SOA
estimated in global sites, respectively; $C_{\text{in-cVMS}}$ and $\Delta C_{\text{cVMS}}$ are the mass concentration of initial and lost
cVMS at the selected equivalent days during the experiments of this work, respectively; $Y$ is the cVMS SOA
mass yields mentioned above. Table S3 summarizes the details regarding sites and concentrations of cVMS
SOA. The derived concentrations of cVMS SOA varies significantly among urban, background and polar
sites. The total cVMS SOA concentrations in urban areas are the highest, up to 1249 ng/m$^3$. They are 14.5-
347 and 10.5-105.3 ng/m$^3$ for background and Arctic sites, respectively. cVMS SOA concentrations in urban
regions of Asia (sites 29-36) and Europe (sites 26-28) are generally larger than that of North America (sites
20-25). In China, the total cVMS SOA concentrations in urban sites range from 14.7 to 1249 ng/m$^3$. The
main precursors of cVMS SOA are different among Chinese cities. For three cities along the southeast coast
of China (Guangzhou, Macau and Foshan), the dominant precursors of cVMS SOA are D3 and D4, which
are related to industrial emissions of these two siloxanes in this region (Wang et al., 2001). For Dalian in
China, mainly D5 and D6 contribute to cVMS SOA, which have the highest concentrations among all sites.
This can be attributed to the industrial production of D5 and D6 and the use of personal care products in
Dalian (Li et al., 2020). In the other Chinese urban areas with reported cVMS concentrations (Lhasa, Golmud,
Kunming and Yantai), the total cVMS SOA concentrations are considerably smaller than those in the urban
areas above, with D5 acting as the main precursor, which may be ascribed to the relatively low population
densities in these cities (Wang et al., 2018).





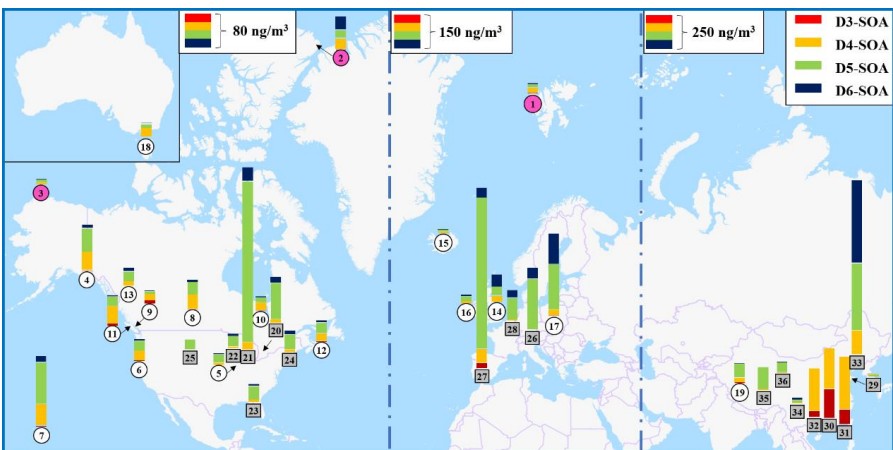


Figure 4. Global concentrations of cVMS SOA (ng/m³) on the basis of the cVMS concentrations
reported from multiple studies and the cVMS SOA yields measured in this work. The numbers of polar,
background and urban sites are enclosed in pink, white circles and gray boxes, respectively. The details
about cVMS and SOA concentrations at different sites were summarized in Table S3 of Supplement.

At urban sites in Europe and North America, cVMS SOA concentrations are reported in the range of 131-
773 ng/m³ and 21.8-416 ng/m³, respectively. Among these cVMS, D5 is the main contributor to cVMS SOA
at these locations, averaging 80.4% of total cVMS SOA. This contribution is higher than that (73.9%) at
Chinese urban sites. For instance, D5 SOA is calculated to be 652 ng/m³ in Catalan, Spain, 369 ng/m³ in
Chicago, USA, 218 ng/m³ in Zurich, Switzerland and 94 ng/m³ in Paris, France, where there are high levels
of economic activities and high population densities. These results suggest that personal care products as a
main source of D5 may be the most important anthropogenic origins of Si-containing SOA in Europe and
North America.
At background and Arctic sites, cVMS SOA are primarily derived from D4 and D5. The background sites
are located in mountains, rural areas, forested areas, lakes and at high altitudes. The three highest cVMS



SOA concentrations at background sites are located at Kosetice in the Czech Republic (347 ng/m$^3$), Hilo,
Hawaii, USA (157 ng/m$^3$) and Tibetan Plateau in China (134 ng/m$^3$), where the contribution percentages of
SOA from both D4 and D5 are 61.9%, 90.3% and 94.9%, respectively. The cVMS SOA concentrations at
the Little Fox Lake site in Yukon, Canada is the highest (105.3 ng/m$^3$) among the four locations in the Arctic,
91% of which is accounted for by both D4 and D5 SOA. The dominance of D4 and D5 SOA in both
background and the Arctic regions highlights their persistence in atmosphere and the potential for long-range
atmospheric transport.

372       Furthermore, global concentration distribution of Si-containing SOA estimated for the four cVMS (D3-

D6) at 36 sites worldwide is also presented in Table S3, which clearly shows the global importance of Si in
SOA with the estimated percentages of cVMS SOA that contains the Si element. For example, up to 49.6%
and 31.2% of cVMS SOA contained Si elements in background and urban sites, respectively. These results
are similar to the summary observations that reported percentages of aerosols with a Si mole fraction >0.01
(%) at different sites (Bzdek et al., 2014). The above results demonstrated that Si is a frequent component of
SOA in background and urban areas.

379       The global annual production of D4, D5 and D6 is about 1, 0.1 and 0.01 Tg·yr$^{-1}$, respectively, and 90% of

these cVMS is eventually released into the atmosphere (Li et al., 2020; Genualdi et al., 2011; Wang et al.,
2013; Sakurai et al., 2019). Based on the results shown in Fig. 4, the annual production of cVMS (D4-D6)
SOA was estimated to be 0.16 Tg·yr$^{-1}$, which was about 5.5% of SOA (2.9 Tg·yr$^{-1}$) produced from mobile
source emissions in the USA and 5-8 times of SOA generated by Athabasca oil sands (0.02-0.03 Tg·yr$^{-1}$, one
of the largest sources of anthropogenic secondary organic aerosols in North America) (Tkacik et al., 2014;
Liggio et al., 2016). Moreover, it was also 0.8% and 2.3% of isoprene-SOA (20 Tg·yr$^{-1}$) and monoterpenes-
SOA (7 Tg·yr$^{-1}$) (typical biogenic SOA), respectively, indicating the potential importance of cVMS SOA



(Jokinen et al., 2015). While these cVMS SOA sources may seem small, they can make substantially higher
contributions to ambient air SOAs in population centers where cVMS compounds are primarily used.

**Author contributions**
CH designed and conducted all experiments; CH and HY analyzed the data and prepared the paper with
contributions from KL, PL, JL, AL and SML. SML supervised the project.

**Competing interests**
The authors declare that they have no conflict of interest.
**Acknowledgements**
This project was supported by Environment and Climate Change Canada's Climate and Clean Air Program
(CCAP); the National Natural Science Foundation of China (42077198); the LiaoNing Revitalization Talents
Program (XLYC1907185); the Fundamental Research Funds for the Central Universities (N182505040,
N2025011).

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
