# Peer review of "Secondary Organic Aerosols from OH Oxidation of Cyclic Volatile Methyl Siloxanes as an Important"

_Atmospheric Chemistry and Physics, 2021_

## Author Comment (AC1)

Dear Professor Jason Surratt,

We welcome the opportunity to revise and clarify our manuscript for publication in *Atmospheric Chemistry and Physics*. Below is a point-by-point response to the comments of reviewers.

**acp-2021-965-RC1**

This manuscript prepared by Han et al. describes an investigation of secondary organic aerosol (SOA) arising from cyclic volatile methyl siloxanes (cVMs). The SOA yield under a variety of atmospheric conditions was investigated using an oxidative flow tube reactor (OFR). The authors provided detailed discussions on the effect of $NO_x$

concentration, presence of seed, and oxidation lifetime. Chemical composition of the resulting SOA was monitored with an aerosol mass spectrometer (AMS). Further, the authors applied the obtained yield in a model to estimate the contribution of cVMS

SOA on the global scale. The scope of the study matches that of ACP. The discussion and conclusions of the paper were fully justified, and the literary quality of the manuscript is outstanding. I highly recommend publication in ACP. My comments below can be considered only technical.

**Re:** We thank you for the positive comments.

One thing that I feel is missing from the current manuscript is a brief discussion on the mechanism of OH reaction with cVMS. This shouldn't take more than a few sentences or a short paragraph. In addition to H-abstraction on the $-CH_3$ groups, does OH undergo any reaction unique to e.g., O-Si bonds? Pieces of mechanistic information are provided in line 265 - 268, but it is hard for the readers to gauge whether that is generic OH

chemistry or something unique to cVMS.

**Re:** The initial step of OH radicals oxidation can be considered as OH abstracting a H

atom from the methyl groups on the -Si-O- ring of cVMS to form Si-containing radicals, which may generate OH and $CH_2OH$ substitution products, such as silanol and silyl methanol (Wu and Johnston, *J. Am. Soc. Mass Spectrom*., 2016, 27, 402-409; Alton and

Browne, *Environ. Sci. Technol*., 2020, 54, 5992-5999). Such Si-containing products may partition into SOA and result in an appearance of $C_xH_yO_zSi_n$ ions in the AMS mass spectra, thereby decreasing the ratio of Si/O (D5: 0.53; D6: 0.72) at initial stage.

Notably, one previous study reported that one of oxidation products of D5, 1- hydroxynonamethylcyclopentasiloxane ($D_4$TOH), has been detected in ambient particulates (Milani et al., *Atmos. Environ.*, 2021, 246, 118078). As the reaction progressed, the Si-O bonds may be cleaved from OH radical attack, which may reduce the number of O atoms, leading to an increase of Si/O with PA. To clearly explain the reaction mechanism of OH with cVMS, the related description has been revised in

Lines 281-289 in the revised manuscript.

Figure S6b - It is a little counterintuitive that the Si/O ratio continued to rise during photooxidation. The number of Si in cVMS is fixed, while the molecule should be continuously oxidized. So Si/O should be decreasing as oxidation proceeds? The authors' discussion from line 259 to 270 did not really explain why Si/O was rising continuously.

**Re:** The generation of OH and $CH_2OH$ substitution products, such as silanol and silyl methanol, may partition into SOA and result in an appearance of $C_xH_yO_zSi_n$ ions in the

AMS mass spectra, thereby decreasing the ratio of Si/O (D5: 0.53; D6: 0.72) at initial stage (Wu and Johnston, *J. Am. Soc. Mass Spectrom.*, 2016, 27, 402-409; Alton and

Browne, *Environ. Sci. Technol.*, 2020, 54, 5992-5999). As the reaction progressed, the

Si-O bonds may be cleaved from OH radical attack, which may reduce the number of

O atoms, leading to an increase of Si/O with PA. The related explanation has been given in Lines 283-289 in the revised manuscript.

Line 151 - the acronym, PTR-ToF-MS is already introduced previously.

**Re:** The acronym of PTR-ToF-MS has been used in Line 156 in the revised manuscript.

**acp-2021-965-RC2**

**Major Comment**

The study was conducted to measure the formation of secondary organic aerosols (SOAs) during OH oxidation of cyclic volatile methylsiloxanes (cVMS) in an oxidation flow reactor (OFR). While the measured data of SOA formation was consistent with other published results obtained similar OFRs, the authors assumed that the results of

OFRs would be used to determine the SOA formation in the real atmosphere. However, the assumption was proved incorrect by an earlier study by Charan et al. (2021). Their work was cited in Introduction of the current manuscript, but the main result of the paper was completely ignored (not mentioned/discussed). Charan et al. (2021)

supported that the yield of SOA depended strongly on OH concentration and organic aerosol mass concentration (related to initial D5 concentration) among several experimental parameters (e.g., OH concentration, OH exposure, aerosol seed, NOx, initial D5 concentration, reactor type). They also claimed that the formation of secondary aerosols is fairly small (<5% or more likely <1%) under environmentally relevant conditions (i.e., low OH and VMS concentrations). Thus, a large chamber might be a good option (rather than OFRs) to maintain environmental-relevant conditions and a low surface-to-volume ratio. Note that all the SOA yields from OFRs were high and in contrast with those from chamber reactors. However, the current manuscript (by Han et al.) did not discussed any of these important findings. They asserted SOA yield was dependent on OH exposure and applied the OFR results for the real atmosphere. Therefore, without providing a strong result against the conclusions of Charan et al. (2021), the current manuscript was built on an incorrect assumption.

Accordingly, the estimated SOA production rate is not reliable.

**Re:** We welcome the opportunity to clarify our paper in this discussion, and respectfully disagree with the comment that "the estimated SOA production rate is not reliable".

This comment concerns the suitability of OFR versus chamber for investigating the yields from cVMS oxidation. This is indeed a generally important discussion topic in the community, and the differences between OFRs versus chambers have been the subject of numerous investigations prior to our paper, with varied conclusions with no clear bias towards the relative merits of either approach. But one clear advantage of

OFRs is their ability to simulate long OH exposures. We are of the opinion that chamber studies generally have difficulties sustaining long OH exposure times, and as such
OFRs are more suitable to address such conditions especially when a compound has
relatively long-life time in the atmosphere. In the case of cVMS, low OH exposure
versus relatively short period of reaction times, such as in the chamber, may indeed
result in a relatively low aerosol yield. However, given the relatively long cVMS life
times in the atmosphere and the continued reaction beyond the chamber reaction times,
one expects continued aerosol production still proceeding that will result in increased
accumulative aerosol yield compared to chamber studies. Such a condition can be
experimentally simulated using OFRs but with difficulties using chambers.

This consideration notwithstanding, it should be noted that the ***published version*** of
the Charan et al. paper (Charan et al., *Atmos. Chem. Phys*., 2022, 22, 917-928) indicated
that at similar OH concentrations and OH exposures, the D5 SOA yields in the Caltech
Photooxidation Flow Tube (CPOT) and chamber experiments were consistent. There
were previous reports for investigating the differences between chambers and flow
reactors. For example, Lambe et al. (*Atmos. Chem. Phys.*, 2015, 15, 3063-3075) found
small differences between chambers and flow reactors for many systems. Peng et al.
(*Atmos. Chem. Phys.*, 2019, 19, 813-834) studied these two types of reactors using
models, and provided operational recommendations for evaluating the atmospheric
relevance of $RO_2$ chemistry. Our previous study (Li et al., *Atmos. Chem. Phys.*, 2019,
19, 9715-9731) made a comparison of SOA yields obtained in ECCC-OFR and other
traditional chambers under similar OH exposure, showing that the SOA yields obtained
by two types of reactors were in good agreement. Under a low OH exposure ($\sim 10^{10}$-
$10^{11}$ molecules $cm^{-3}$ s), the D5 SOA yield (0.01-0.11) obtained here was similar to that
(chamber, 0-0.057; flow tube, 0.018-0.06) measured by Charan et al. (*Atmos. Chem.*
*Phys*., 2022, 22, 917-928).

Our differences with Charan et al. (*Atmos. Chem. Phys*., 2022, 22, 917-928) are for
high OH exposure conditions both using OFRs. Under a high OH exposure of $\sim 10^{11}$-
$10^{12}$ molecules $cm^{-3}$ s, the D5 SOA yield of 0.46-0.70 was higher than 0.14-0.35 (flow
tube) reported by Charan et al. (*Atmos. Chem. Phys*., 2022, 22, 917-928). Here the
differences may be attributed to differences in experimental conditions, such as differences in reactors, wall losses, SOA measurement methods, determination of OH concentrations, and initial D5 concentrations (Table S5) (Janechek et al., *Atmos. Chem. Phys.*, 2019, 19, 1649-1664; Charan et al., *Atmos. Chem. Phys.*, 2022, 22, 917-928). Wall losses in particular can be a major factor influencing aerosol yields; our ECCC-OFR uses a sheath flow to minimize such an artefact (Li et al., *Atmos. Chem. Phys.*, 2019, 19, 9715-9731) resulting in higher and more reliable yields than approaches without such a design. In the revision, we have now included a brief discussion to include the results from the Charan et al. paper in Lines 200-208, and further in Table S5 in the revised *Supplement* to show the comparison.

When extrapolating laboratory results to the real atmosphere, the relevant OH concentrations and exposures should be understood carefully, as suggested by Charan et al. (*Atmos. Chem. Phys.*, 2022, 22, 917-928). As we all know, one important advantage of OFR is to simulate up to several weeks of OH exposures. Considering the half-life of cVMS, the OH exposures on the order of $10^{12}$ molecules cm$^{-3}$ s are also environmentally relevant (Charan et al., *Atmos. Chem. Phys.*, 2022, 22, 917-928). Studies have predicted the SOA formation based on ambient SOA precursors and yields obtained from OFR (Palm et al., *Atmos. Chem. Phys.*, 2018, 18, 467-493; Jokinen et al., *Proc. Natl. Acad. Sci. U S A.*, 2015, 112, 7123-7128), further strengthening the argument in favor of OFRs.

The calculation of cVMS-SOA concentration (in Eq. 4) is not clear. The equation needs the amount of lost of cVMS, but the quantity was not measured or published. See more in Minor Comment below.

**Re:** The reacted cVMS in the atmospheric environment is estimated from $C_{cVMS} \times \Delta C_{cVMS}/C_{in\text{-}cVMS}$, based on assumptions that the lost cVMS ratio is not affected by the cVMS concentration and the background $C_{cVMS\text{-}SOA}$ is zero. The values of lost cVMS ($\Delta C_{cVMS}$) were obtained by the PTR-MS measurements, and they have been provided in Table S7 (i.e., Table R2) in the revised *Supplement*.

The manuscript states that the number and mass size distribution of aerosols was

149 monitored using a scanning mobility particle sizer (SMPS), but no data was provided

150 and discussed except the statement that the data was used for calculation of effective

151 aerosol densities. The results would be helpful to understand the characteristics of SOA

152 formation under different conditions.

**Re:** Based on the SPMS data, the number and mass size distributions of aerosols (D5

SOA as an example) were added in Figure S2 (i.e., Figure R1) in the revised *Supplement*.

Figure S2 (i.e., Figure R1) shows that small particles dominated in the number, while large ones were prominent in the mass. Compared to the size distributions of D5 SOA

in low-NO$_x$ experiments (Figure. S2a), the mode diameters of SOA increased for both number and mass size distributions under high-NO$_x$ conditions (Figure. S2b), which suggests that larger particles are dominant in D5 SOA generated in high-NO$_x$

experiments. The related discussions have been added in Lines 175-180 in the revised manuscript.

[Figure]

Figure R1. Number and mass size distributions of D5 SOA measured by SMPS at OH

exposure of $9.0 \times 10^{11}$ molecules cm$^{-3}$ s under low-NO$_x$ (**a**) and high-NO$_x$ (**b**) conditions.

**Minor Comment**

168 L51-53: Is this (scarce knowledge) still valid? There have been many studies published

169 since the reference you cited.

**Re:** It has been modified in Lines 51-53 in the revised manuscript as shown below:

"The legislative actions notwithstanding, knowledge of environmental behavior of cVMS still needs to be further deepened as compared to their applications and economic significance".

**Re:** The initial concentrations for each of the cVMS species have been provided in

Table S2 (i.e., Table R1) in the revised *Supplement*. It should be noted that these were far greater than environmental concentrations, which may cause potential uncertainty in the extrapolation to the atmospheric conditions. This has been described in Lines

356-360 in the revised manuscript.

              Table R1. Initial concentration of cVMS

| cVMS | Low-NO$_x$ experiments (ppb) | | High-NO$_x$ experiments (ppb) | |
|------|----------|--------|----------|--------|
|      | unseeded | seeded | unseeded | seeded |
| D3   | 19-21    | 18-20  | 38-43    | 42-46  |
| D4   | 27-35    | 30-33  | 25-31    | 25-33  |
| D5   | 18-26    | 27-30  | 36-41    | 30-38  |
| D6   | 19-28    | 25-30  | 24-30    | 26-32  |

L190 (&L29): This study claims that the production of SOA is dependent on OH

exposure. However, this is partially true in the limit of OFR experiments. In contrast,

Charan et al. (2021) discovered that OH radical concentrations are more important than

OH exposure considering all experiment results with OFR and chamber reactors. If the

latter is valid, it's better to use OH radical concentrations for the x-axis rather than

photochemical age as the initial cVMS concentrations were in a relatively narrow range.

**Re:** It should be pointed out that the comparison on how OH concentration versus exposure affects SOA yields has been removed in the published version of Charan et al. (*Atmos. Chem. Phys.*, 2022, 22, 917-928). In fact, the OH exposure (i.e., photochemical age) is positively correlated with the OH concentration due to the constant resident time of gas in the ECCC-OFR. The use of OH exposure (i.e., photochemical age) is a typical way for OFR experiments (Tengyu Liu et al., *Atmos. Chem. Phys.*, 2018, 18, 5677-5689; Janechek et al., *Atmos. Chem. Phys.*, 2019, 19, 1649-1664). As for us, it is better to use OH exposure as the x-axis, which is convenient to compare with results from other literatures.

L255-256: This is under unseeded conditions. Please explain why there were more productions under seeded conditions at earlier PAs?

**Re:** As shown in Figure 3, the initial fractions of $C_xH_y$ in seeded experiments were larger than those in unseeded experiments, which may be related to the volatility of species containing $C_xH_y$ that may be more easily deposited in the presence of seeds. This has been given in Lines 300-301 in the revised manuscript.

L321-322: The estimation would not be valid based on this study that used OFRs. See Charan et al. (2021) for the detailed discussion.

**Re:** The responses to the first major comment have clarified the validity of the estimation. It should be noted that in the recently ***published version*** of Charan et al. (*Atmos. Chem. Phys.*, 2022, 22, 917-928) the CPOT OFR was also used to support their conclusion. Briefly, when extrapolating laboratory results to real atmosphere, the relevant OH concentrations and exposures should be understood carefully, as suggested by Charan et al. (*Atmos. Chem. Phys.*, 2022, 22, 917-928). As we all know, one important advantage of OFR is to simulate up to several weeks of OH exposure. Considering the half-life of cVMS, the OH exposures on the order of $10^{12}$ molecules cm$^{-3}$ s are also environmentally relevant (Charan et al., *Atmos. Chem. Phys.*, 2022, 22, 917-928). Additionally, previous studies have predicted the SOA formation based on ambient SOA precursors and yields obtained from OFR (Palm et al., *Atmos. Chem. Phys.*, 2018, 18, 467-493; Jokinen et al., *Proc. Natl. Acad. Sci. U S A.*, 2015, 112, 7123-

7128), further demonstrating the usefulness of OFR in investigating reactions relevant to the atmosphere.

L323,330: There should be differences between $C_{cVMS}$ and $\Delta C_{cVMS}$. I assume SOA

concentrations were calculated based on $C_{cVMS}$, not $\Delta C_{cVMS}$. Eq.4 is valid only when

$C_{cVMS} = C_{in\text{-}cVMS}$ and background $C_{cVMS\text{-}SOA} = 0$. Thus, there should be more clarification.

In addition, extrapolation should be highly uncertain due to 4 orders-of-magnitude differences of concentrations and overestimated SOA yields.

**Re:** The reacted cVMS in the atmospheric environment is estimated from

$C_{cVMS} \times \Delta C_{cVMS} / C_{in\text{-}cVMS}$, based on assumptions that the lost cVMS ratio is not affected by the cVMS concentrations and the background $C_{cVMS\text{-}SOA}$ is zero. Although there may be some uncertainties in extrapolating our results to the real atmosphere, such as larger cVMS concentrations and SOA yields, our extrapolations may provide an upper limit for evaluating cVMS SOA. This has been described in Lines 356-360 in the revised manuscript.

L332-333: Not clear what values were used in this study. The values were not directly measured.

**Re:** The values of $\Delta C_{cVMS} / C_{in\text{-}cVMS}$ were obtained by the PTR-MS measurements, and they have been provided in Table S7 (i.e., Table R2) in the revised *Supplement*.

Table R2. The values of $C_{in\text{-}cVMS}$ and $\Delta C_{cVMS}$ in Equation 4.

| cVMS | Low-NO$_x$-unseeded-14.2 d | | High-NO$_x$-seeded-8.5 d | |
|------|---------------------------------------------------------|-----------------------------------------------------|----------------------------------------------------|----------------------------------------------------|
|      | Initial concentration ($C_{in\text{-}cVMS}$, µg m$^{-3}$) | Lost concentration ($\Delta C_{cVMS}$, µg m$^{-3}$) | Initial concentration ($C_{in\text{-}cVMS}$, µg m$^{-3}$) | Lost concentration ($\Delta C_{cVMS}$, µg m$^{-3}$) |
| D3   | 186.83 | 156.99 | 392.56 | 258.27 |
| D4   | 386.42 | 327.30 | 391.84 | 305.99 |
| D5   | 371.66 | 371.66 | 557.73 | 445.59 |
| D6   | 476.12 | 453.72 | 566.47 | 456.72 |

**Re:** It has been added in Line 51 in the revised manuscript.

**acp-2021-965-RC3**

The manuscript by Han et al describes several cVMS aerosol yield experiments in an oxidation flow reactor (OFR). They use their derived yields to make estimates of cVMS-derived SOA at various locations. cVMS chemistry and aerosol yield is an open question in the field and the results are of interest to the community. However, I have several major comments that I would like to see addressed before I could recommend publication.

**Re:** We welcome the opportunity to further clarify and improve our paper.

**Major Comments**

The results of this study need to be discussed in the context of Charan et al., (2021). Charan et al. (2021) found that the aerosol yield for D5 oxidation varied with the OH exposure in an unusual way. I note that when considering the Charan et al., (2021) work, it would be prudent to wait for the ACP version of the manuscript as several points of discussion regarding possible explanations of the differing yields were modified following public discussion. A particular result from Charan et al. (2021) that should be discussed in the context of the current study is that OFR experiments led to higher SOA yields than environmental chamber experiments.

**Re:** The ***published version*** of Charan et al. (*Atmos. Chem. Phys*., 2022, 22, 917-928) showed that at similar OH concentrations and OH exposures, the D5 SOA yields in the Caltech Photooxidation Flow Tube (CPOT) and chamber experiments were consistent. Under a low OH exposure ($\sim 10^{10}$-$10^{11}$ molecules cm$^{-3}$ s), the D5 SOA yield (0.01-0.11) obtained here was similar to that (chamber, 0-0.057; flow tube, 0.018-0.06) measured by Charan et al. (*Atmos. Chem. Phys*., 2022, 22, 917-928). However, under a high OH exposure of $\sim 10^{11}$-$10^{12}$ molecules cm$^{-3}$ s, the D5 SOA yield of 0.46-0.70 was higher than 0.14-0.35 (flow tube) reported by Charan et al. (*Atmos. Chem. Phys*., 2022,

22, 917-928), respectively, which may be attributed to differences in experimental conditions, such as differences in reactors, wall losses, SOA measurement methods, determination of OH concentrations, and initial D5 concentrations (Table S5)

(Janechek et al., *Atmos. Chem. Phys.*, 2019, 19, 1649-1664; Charan et al., *Atmos. Chem.*

*Phys*., 2022, 22, 917-928). The results from Charan et al. (*Atmos. Chem. Phys*., 2022,

22, 917-928) have been cited in Table S5 in the revised *Supplement.* Moreover, a brief discussion has been added in Lines 200-208 in the revised manuscript.

Further details about the radical chemistry need to be included in the manuscript. While information is given on the ratio or $RO_2 + NO$ to $RO_2 + HO_2$ reactions in the high $NO_x$

experiments, there is no discussion on how prevalent $RO_2 + RO_2$ reactions were or even the (not atmospherically relevant) $RO_2 + OH$. Differences in the fate of the peroxy radical in the OFR experiments compared to chamber experiments is one possibility that could explain the Charan et al (2021) results, however, the investigation of $RO_2$

fate was beyond the scope of that paper. I encourage the authors to describe the radical chemistry of these experiments in more detail as it could aid in understanding the apparent discrepancy between OFR experiments and chambers for this chemical system.

**Re:** Although $RO_2+RO_2$ reactions cannot be totally ignored compared to $RO_2 + HO_2$

and $RO_2 + NO$ reactions, their roles may be minor due to the low concentration of SOA

precursors (cVMS, 18-46 ppb) according to the study of Lambe et al. (*Atmos. Meas.*

*Tech.*, 2017, 10, 2283-2298). Peng et al. (*Atmos. Chem. Phys.*, 2019, 19, 813-834) have illustrated that when at low precursor concentrations, the roles of $RO_2 + RO_2$ are minor or negligible in OFR, where $N_2O$ was used to achieve high-$NO_x$ conditions. Our previous model results have also demonstrated that the dominant fate of $RO_2$ was the reaction with NO in the ECCC-OFR under high-$NO_x$ conditions (Li et al., *Environ. Sci.*

*Technol.*, 2019, 53, 14420-14429), implying the minimal role of $RO_2 + RO_2$ reactions.

As reported by Peng et al. (*Atmos. Chem. Phys.*, 2019, 19, 813-834), the relative importance of $RO_2 + OH$ is generally negatively correlated with the injection of $N_2O$

in OFR, which may be attributed to the suppressing effect of $NO_x$ on OH and the increasing role of $RO_2$ + NO. The related discussion about the radical chemistry has been added in Lines 147-152 in the revised manuscript.

Given the prevalence of cVMS, it is easy for contamination to occur. I would like to see more information on control experiments that were performed to check for contamination and possible impacts on cVMS yields. Additionally, I think the manuscript would benefit from including more examples (SI is ok) of the measurements. In particular, I would like to see some examples of the SMPS data and AMS time series absolute values (not in terms of fraction). The SMPS data in particular should be included as it is central in the yield calculation.

**Re:** Control experiments were performed under both low and high-$NO_x$ conditions, where the aerosol mass concentration (background values) in the absence of cVMS was measured by SMPS, as shown in Table S3 (i.e., Table R3) in the revised *Supplement*. The background values were subtracted when calculating the cVMS SOA yields. The revised descriptions have been added in Lines 171-172 in the revised manuscript.

Taking D5 in low-$NO_x$ experiments as an example, the mass concentrations of D5 SOA measured by SMPS under unseeded conditions were shown in Table S4 (i.e., Table R4) in the revised *Supplement*. Moreover, the number and mass size distributions of D5 SOA measured by SMPS were added in Figure S2 (i.e., Figure R1) in the revised *Supplement*. The related discussions have been added in Lines 175-180. The AMS time series of D5 SOA in low-$NO_x$ experiments under unseeded conditions were also shown in Figure S3 (i.e., Figure R2) in the revised *Supplement*. This has been described in Lines 180-182 in the revised manuscript.

Table R3. Background values of SMPS at different OH exposures in low and high-$NO_x$ experiments without cVMS.

| Low-$NO_x$ experiments | | | High-$NO_x$ experiments | | | |
|---|---|---|---|---|---|---|
| OH exposure | Equivalent photochemic | Backgroun d ($\mu g/m^3$) | OH exposure | Equivalent photochemic | Backgroun d ($\mu g/m^3$)- | Backgroun d ($\mu g/m^3$)- |

| (×10$^{12}$ molecules cm$^{-3}$ s) | al age (days) | | (×10$^{12}$ molecules cm$^{-3}$ s) | al age (days) | unseeded | seeded |
|---|---|---|---|---|---|---|
| 0.05 | 0.42 | 0.51 | 0.08 | 0.63 | 0.15 | 2.74 |
| 0.14 | 1.10 | 1.55 | 0.22 | 1.71 | 0.33 | 5.37 |
| 0.26 | 2.00 | 2.01 | 0.34 | 2.64 | 0.32 | 6.84 |
| 0.35 | 2.67 | 2.38 | 0.45 | 3.45 | 0.39 | 7.44 |
| 0.43 | 3.28 | 2.25 | 0.61 | 4.69 | 0.37 | 10.91 |
| 0.50 | 3.86 | 2.70 | 0.82 | 6.36 | 0.58 | 12.34 |
| 0.60 | 4.66 | 3.50 | 0.90 | 6.94 | 0.41 | 12.21 |
| 0.69 | 5.30 | 3.73 | 1.10 | 8.46 | 0.98 | 13.61 |
| 0.90 | 6.95 | 3.86 | | | | |
| 1.16 | 8.96 | 5.19 | | | | |
| 1.32 | 10.15 | 6.17 | | | | |
| 1.67 | 12.88 | 7.24 | | | | |
| 1.85 | 14.24 | 7.81 | | | | |

Table R4. The mass concentrations of D5 SOA at different OH exposures under unseeded conditions in low-NO$_x$ experiments.

| OH exposure (×10$^{12}$ molecules cm$^{-3}$ s) | Equivalent photochemical age (days) | Low-NO$_x$-unseeded-D5 SOA (μg/m$^3$) |
|---|---|---|
| 0.05 | 0.42 | 0.49 |
| 0.14 | 1.10 | 1.83 |
| 0.26 | 2.00 | 8.92 |
| 0.35 | 2.67 | 16.95 |
| 0.43 | 3.28 | 25.76 |
| 0.50 | 3.86 | 48.86 |
| 0.60 | 4.66 | 68.65 |

| 0.69 | 5.30 | 97.70 |
|------|------|-------|
| 0.90 | 6.95 | 169.74 |
| 1.16 | 8.96 | 228.76 |
| 1.32 | 10.15 | 253.56 |
| 1.67 | 12.88 | 282.73 |
| 1.85 | 14.24 | 273.61 |

[Figure]

Figure R1. Number and mass size distributions of D5 SOA measured by SMPS at OH

exposure of $9.0 \times 10^{11}$ molecules cm$^{-3}$ s under low-NO$_x$ (**a**) and high-NO$_x$ (**b**)

conditions.

[Figure]

Figure R2. The AMS time series of D5 SOA in low-NO$_x$ experiments under unseeded conditions. The numbers represent the equivalent photochemical age.

The back of the envelope calculations for the cVMS SOA loadings need to be better

justified. For instance, why is 8.5 days selected for the urban sites? That seems long considering typical transport times. There should also be discussion of the results of

Milani et al., (2021) and Pennington et al., (2021) and the consistency or not of these back of the envelope estimates with those studies.

**Re:** The atmospheric half-lives of D3-D6 were 6-30 days. The yield at 8.5 equivalent days was selected to provide an upper limit for evaluating cVMS SOA. This has been described in Lines 349-350 and 356-360 in the revised manuscript.

A discussion with the results of Milani et al. (*Atmos. Environ.*, 2021, 246, 118078)

and Pennington et al. (*Atmos. Chem. Phys.*, 2021, 21, 18247-18261) has been added in

Lines 388-394 in the revised manuscript and presented below.

"It was noted that the D5 SOA concentration (13.38-683.57 $ng/m^3$) estimated here is far more than that (0.016-0.206 $ng/m^3$) reported by Milani et al., (2021), who obtained their data using semi-quantified concentrations of $D_4TOH$ (first-generation D5 SOA

product) extracted from $PM_{2.5}$ samples in Atlanta and Houston. The difference may be mainly attributed to the missing analysis of multi-generation SOA products or dimers (Wu and Johnston, 2016, 2017). Pennington et al., (2021) utilized the developed CMAQ

model to investigate the concentration of D5 SOA in the urban area of Los Angeles, and the model data (21 $ng/m^3$) was within these values here."

**Minor Comments**

Figure 2 (and analogous SI figure for high $NO_x$). Given that the cVMS precursors only differ in the number of $(C_2H_6OSi)$ units, I would have expected to see somewhat more similar mass spectra from their oxidation products. I find it intriguing how different the

MS are between the experiments and find some of the ions unexpected. For instance,

$C_{12}H_{11}O_2Si_5^+$ from D5 oxidation is intriguing given the addition of C and loss of O from the parent species. I recognize the electron ionization spectra of complex mixtures can be challenging and non-intuitive to interpret, but I wonder if the authors have considered possible explanations for some of these ions and what they might imply for the identify of oxidation products. Additionally, I wonder if the PTR provides any information on gas-phase oxidation products that is useful.

**Re:** We thank the reviewer for this thoughtful comment. Indeed, information buried in the MS can be very meaningful for understanding the mechanisms of cVMS oxidation by OH radicals. We have tried to provide possible explanations for these ions when analyzing mass spectra. However, due to the complexity of the fragments caused by electron ionization, we admit that it is hard to establish a correlation between these special fragments and oxidation products only based on the AMS results. Gas-phase species in PTR-MS are still being analyzed in our further investigation to complete a study about the kinetics and gas-phase products of the reaction between cVMS and OH

radicals. The PTR-MS may provide some information about the gas-phase species, as shown in Figure R3. The focus of the current manuscript is on the yields and compositions of cVMS SOA, rather than gas-phase products and mechanisms of cVMS

oxidated by OH radicals, which are the subject of a continued investigation. Therefore, the PTR-MS results are not presented in our manuscript.

[Figure]

Figure R3. The mass spectrum of the gas-phase oxidation products of D5 measured by PTR-MS under low-$NO_x$ and unseeded conditions.

Lines 268-270: I would think that continued breaking of Si-O bonds would lead to more volatile products since after the first bond breaking, it would lead to fragmentation and thus smaller molecules.

**Re:** We agree with your opinion. The related explanation has been modified in Lines

290-291 in the revised manuscript.

**Technical Comments**

Line 70: The conclusion of Alton and Browne, (2020) is more subtle than suggested here. They suggest that while Cl is minor globally, loss locally (for instance in Los Angeles) could be more important given the spatial and temporal overlap of cVMS emissions and sources of Cl (i.e. from $ClNO_2$). This is an important distinction if cVMS yields are high as suggested by this work.

**Re:** The text has been modified in Lines 70-71 in the revised manuscript and shown below: "The loss of cVMS in the atmosphere is slight through $O_3$ and $NO_3$ due to their small reaction rates, and by Cl atoms on account of its low concentration except for the spatial and temporal overlaps of cVMS emissions and Cl sources."

Lines 84-85: "…indicating necessary conditions when extrapolating SOA yields to the ambient atmosphere." This sentence does not make sense. Pleas reword.

**Re:** This sentence has been rewritten in Lines 84-86 in the manuscript as "…emphasizing the importance of the relevant OH concentrations and exposures when extrapolating these laboratory results or comparing with other studies…".

Table S2: Please keep D5 in the same units for all the studies.

**Re:** The unit of D5 in Table S5 has been uniformly changed to "ppbv" in the revised *Supplement*.

Line 229: Alton and Browne (2020) did not state anything about the relative volatility of different cVMS oxidation products. The only discussion on volatility concerned wall loss and that one of the first-generation oxidation products had a lower vapor pressure than the parent cVMS. Moreover, that vapor pressure is still high enough that it is unlikely to contribute to SOA except at high loadings. Additionally, they did not report D6 results.

**Re:** This reference has been deleted in Line 245 in the revised manuscript.

The use of n/z and n/z rather than Si/C and Si/O adds complexity to the paper that I think is unnecessary. I think it would be easier for the reader to understand the results if Si/C and Si/O are used.

**Re:** "Si/C" and "Si/O" have been used in Lines 276-280 and 331-333 in the revised manuscript.

Lines 265-266: This discussion of oxidation as occurring by substitution of one functional group for another is confusing and misleading. I suggest editing to be more clear about the chemistry occurring (OH abstracting a H, etc.).

**Re:** This discussion has been modified into "The initial step of OH radical oxidation can be considered as OH abstracting a H atom from the methyl groups on the -Si-O- ring of the cVMS to form Si-containing radicals, which may generate OH and $CH_2OH$ substitution products, such as silanol and silyl methanol (Wu and Johnston, 2016; Alton and Browne, 2020)", as shown in Lines 281-284 in the revised manuscript.

References

Alton, M. W. and Browne, E. C.: Atmospheric Chemistry of Volatile Methyl Siloxanes: Kinetics and Products of Oxidation by OH Radicals and Cl Atoms, Environ. Sci. Technol.,

54, 5992–5999, https://doi.org/10.1021/acs.est.0c01368, 2020.

Charan, S. M., Huang, Y., Buenconsejo, R. S., Li, Q., Cocker III, D. R., and Seinfeld, J. H.: Secondary Organic Aerosol Formation from the Oxidation of Decamethylcyclopentasiloxane at Atmospherically Relevant OH Concentrations, Atmospheric Chem. Phys. Discuss., 1–17, https://doi.org/10.5194/acp-2021-353, 2021.

Milani, A., Al-Naiema, I. M., and Stone, E. A.: Detection of a secondary organic aerosol tracer derived from personal care products, Atmos. Environ., 246, 118078, https://doi.org/10.1016/j.atmosenv.2020.118078, 2021.

Pennington, E. A., Seltzer, K. M., Murphy, B. N., Qin, M., Seinfeld, J. H., and Pye, H. O. T.: Modeling secondary organic aerosol formation from volatile chemical products, Atmospheric Chem. Phys., 21, 18247–18261, https://doi.org/10.5194/acp-21-18247-

2021, 2021.

Re: We thank the reviewer for providing these literatures. We have read them carefully and some of them have been cited and discussed properly in the revised manuscript.

Sincerely yours,

Chong Han, Professor

School of Metallurgy

Northeastern University

Shenyang 110819, China

E-mail: hanch@smm.neu.edu.cn

Shao-Meng Li, Chair Professor

College of Environmental Sciences and Engineering

Peking University

Beijing, China 100871

E-mail: shaomeng.li@pku.edu.cn

---

## Author Response (AR2)

Dear Professor Jason Surratt,

We welcome the opportunity to revise and clarify our manuscript for publication in *Atmospheric Chemistry and Physics*. Below is a point-by-point response to the comments of reviewers.

**Anonymous referee #2**

I understand the experiments with the oxidation flow reactor (OFR) were conducted well. However, I am not fully convinced that the results can be applied to the environmentally relevant conditions. Charan et al. (2022) claimed that the yield of SOA depended strongly on OH concentration, not OH exposure. The OH exposure (OH concentration multiplied by exposure duration) and photochemical age can be used interchangeably using an assumption of a constant OH concentration. OH exposure can be used to determine the degree of degradation of a parent compound. A low OH concentration for a long exposure time and a high OH concentration for a short exposure time can achieve the same degree of degradation of the parent compound. However, that is not for the formation of SOA as Charan et al. (2022) asserted. The current experiments were conducted with a fixed residence time of 2 minutes while OH concentrations were varied. Thus, the main independent variable should be OH concentration, but not OH exposure nor photochemical age.

**Re**: As you suggested, the OH concentration has been used as a variable in Figures 1, 3, S5, S6, S8, S10 and S11 in the revised manuscript and *Supplement*.

In addition, the SOA yields from the experiments cannot be used to predict the formation of SOA for various monitoring sites if the claim of Charan et al. (2022) is accepted. Unless the authors fully discuss why the results from an OFR study can be applied to the environmental conditions, all the predictions of SOA formation in the real world are already unreliable.

**Re**: We agree that OH concentrations and exposures have different effects on the reaction systems, leading to different SOA yields. OH concentration determines the cVMS+OH reaction rate and therefore the instantaneous cVMS SOA yield. In contrast, OH exposure of cVMS determines the time-integrated, or cumulative, cVMS SOA yield. For any OH concentration, if a cVMS is exposed to the OH for a short period of time, the cumulative SOA yield over the short exposure time will be small. From this perspective, exposure is a more relevant factor to atmospheric conditions because SOA yield under such conditions should be an integration of instantaneous yields over the lifetime of a precursor compound in the atmosphere. Hence, one can argue that exposure experiments such as ours, simulating exposure in the real atmosphere, that were conducted using OFR, should be more suitable for application to the ambient atmosphere.

As Charan et al. described, the interchanges between OH concentrations and exposures have to be considered due to experimental limitations, especially an inability to carry out long-time (multiple days) experiments (Charan et al., *Atmos. Chem. Phys.*, 2022, 22, 917-928). When extrapolating the laboratory data to the real atmosphere, it is necessary to consider atmospherically relevant OH concentrations and exposures. Charan et al. have claimed that D5 SOA yields are strongly dependent on both OH concentrations and exposures (*Atmos. Chem. Phys.*, 2022, 22, 917-928), implying the simultaneous effects of OH concentrations and reaction time on SOA yields. OH exposures in our work are $0.05\text{-}1.85\times10^{12}$ molecules $cm^{-3}$ s, which is within atmospheric OH exposure range of cVMS when considering half-lives (6-30 days) of cVMS and average OH concentration ($1.5 \times 10^6$ molecules $cm^{-3}$) in the atmosphere. Therefore, these experimental data is used to predict the SOA formation in real environments, which would provide an estimation for understanding the SOA potential of cVMS. The related descriptions have been stated in Lines 373-392 in the revised manuscript.

I also observed that the concentrations of cVMS at the exit of the reactor were not clearly presented and discussed although they were measured according to the manuscript. The results would indicate how deep reactions went under different conditions.

**Re**: The concentrations of each cVMS at the exit of the reactor have been added in Table S2 (i.e., Table R1) in the revised *Supplement*. The related descriptions have been stated in Lines 140-142 in the revised manuscript.

       Table R1. The concentration of cVMS at the reactor inlet and outlet.

| cVMS | Low-NO$_x$ experiment (ppb) | | High-NO$_x$ experiment (ppb) | |
|---|---|---|---|---|
| | unseeded | seeded | unseeded | seeded |
| $[D3]_i$ | 19.91 | 20.06 | 41.20 | 42.90 |
| $[D3]_o$ | 3.28 | 3.14 | 13.03 | 14.76 |
| $[D4]_i$ | 30.89 | 29.83 | 27.05 | 28.56 |
| $[D4]_o$ | 4.87 | 4.51 | 5.95 | 7.08 |
| $[D5]_i$ | 21.51 | 28.45 | 38.50 | 35.99 |
| $[D5]_o$ | 0 | 0.12 | 7.31 | 7.39 |
| $[D6]_i$ | 24.40 | 27.08 | 26.92 | 28.67 |
| $[D6]_o$ | 1.23 | 1.16 | 5.73 | 6.03 |

Note: 'i' and 'o' means the reactor inlet and outlet, respectively; the OH exposure is $1.85\times10^{12}$

and $1.10\times10^{12}$ molecules cm$^{-3}$ s in low and high-NO$_x$ experiments, respectively.

Another minor additional comment is that the effect of seasons and climate changes on cVMS

reaction should be discussed briefly in Introduction although not extensively since OH radical formation is dependent on solar illumination.

**Re**: The related discussion has been given in Lines 70-72 in the revised manuscript: "Different

OH concentrations can partly explain the seasonal variation of cVMS lifetimes that was characterized by longer during winter than in summer (Rücker and Kümmerer, 2015)".

**Anonymous referee #3**

I thank the authors for their consideration and response to my prior comments. I think that the manuscript is improved; however, I believe it would benefit from further consideration of the points below. Line numbers refer to the track changes version of the manuscript.

**Re**: Thank you for your comments.

**Main comments**

1) Thank you for the additional text describing the peroxy radical fate in the high NO$_x$

conditions. I think that the same consideration should be given to the low $NO_x$ conditions.

Based on line 145, it appears that a model was used to determine the $RO_2$ fate for the high $NO_x$

experiment. Including the general output of that model (for instance, the model estimates over x% of $RO_2$ reacts with $HO_2$) for the low-$NO_x$ simulations would be more convincing to the reader than the current statements.

**Re**: The reaction ratio of $RO_2$ with $HO_2$ was estimated to be larger than 99% under low-$NO_x$

conditions. The related description has been given in Lines 159-161 in the revised manuscript.

2) Lines 177-179 (in reference to the SMPS data in Fig. S2): I don't think this statement is fully supported by the information available to me as a reader. For instance, couldn't the distribution be different because of different starting conditions? From table S2, low $NO_x$ unseeded D5

experiments had 18-26 ppb D5 whereas high-$NO_x$ had 36-41 ppb D5. The different starting concentrations would affect the distributions. With regards to figure S2, please clarify the meaning of "normalized" in the axes. It is not readily apparent to me how these values were normalized. It is also unclear to me how representative these SMPS distributions are.

**Re**: We have presented three sets of SPMS data at low, medium and high photochemical age (PA) in low and high-$NO_x$ experiments, as shown in Figure S3 (i.e., Figure R1) in the revised

*Supplement*. The mass mode diameter of SOA for mass size distributions increased with PA

under low and high-$NO_x$ conditions. The related descriptions have been modified in Lines 184-

188 in the revised manuscript. With regards to Figure S3 (i.e., Figure R1), the data was not normalized, and the word "normalized" in the Y-axes has been deleted.

[Figure]

Figure R1. Number and mass size distributions of D5 SOA from SMPS under (**a**) low-$NO_x$

(PA=2.0, 6.9 and 14.2 days) and (**b**) high-$NO_x$ (PA=1.7, 3.4 and 6.9 days) conditions.

3) Figure S3: I think the data presented in this figure warrants further discussion and a more transparent consideration of how experimental variability/error influences the results.

Particularly at the higher OH exposures, the OA loading doesn't plateau during an experiment, meaning that the yields calculated will be extremely dependent upon which data was used in the calculation. I think some discussion on why the mass loading fails to stabilize under the high OH conditions and how this impacts the derived quantities and interpretation is essential.

Moreover, there should be variability/error bars reported on the values. This discussion and error analysis is particularly important given the discussion in the literature about the sensitivity to this chemical system to high OH exposures (e.g. the Charan paper).

**Re**: The AMS results presented in this work were derived from the average values in the last

10 runs (~10 minutes) for each OH exposure, as shown in the inset of Figure S2 (i.e., Figure

R2) in the revised *Supplement*. At high OH exposures, unstable SOA loadings may be mainly attributed to the fragmentation reactions, leading to the difficulty in the deposition of products on SOA. The yields were calculated from the average data of SMPS and AMS in the last 10

minutes for each OH exposure. The error bars of the SOA yields have been given in Figure 1

in the revised manuscript, and Figure S5 in the revised *Supplement*. The related descriptions have been stated in Lines 179-181 and 190-191 in the revised manuscript.

[Figure]

Figure R2. The AMS time series of D5 SOA in low-$NO_x$ experiments under unseeded conditions. The numbers represent the equivalent photochemical age, and the inset is an enlarged view at 10.2 and 12.9 PA.

4) I am not convinced by the discussion of why the Si/O increases with photochemical age (text added on lines 288-291). The added text is hand-wavy and doesn't explain why the Si/O would increase at the lowest photochemical ages where siloxane loss (and thus generation of higher oxidation products) is low. Furthermore, the Si-O bond is an incredibly strong bond (> 110

kcal/mol). This can be compared to Si-C, C-C, or C-O bonds (all <90 kcal/mol). Bond energies are from (Rücker and Kümmerer, 2015). What support in the literature is there for this idea of

Si-O bond breaking, particularly at the low OH exposures? I wonder what the role of fragmentation in the AMS detection and/or uncertainty in peak assignment could be to the reported Si/O values. Elemental ratios derived from AMS measurements typically require at least some sort of correction. Given the variability in the MS pattern b/w siloxane precursor (as mentioned in my minor comment #1 from the initial review and the associated author response), fragmentation may play a significant role in the observed Si/O. Some further discussion on the uncertainty associated with the Si/O calculation is warranted.

**Re**: At low OH exposure, some oxygen-containing functional groups (-CH$_2$OH/-COOH/-OH)

can be formed by the reaction of methyl groups in cVMS with OH, which resulted in a smaller

Si/O ratio compared to that (1.0) in cVMS. With increasing OH exposure, these functional groups (-CH$_2$OH/-COOH/-OH) may dissociate, leading to an increase in the Si/O ratio. The

Si-O bond breaking may mainly happen at high OH exposures, and it may occur after the cleavage of S-C bonds (69 kcal/mol) (Rücker and Kümmerer, *Chem. Rev.*, 2015, 115, 466-524).

According to the previous study of Wu and Johnston (*Environ. Sci. Technol.*, 2017, 51, 4445-

4451), some ring-opened products were generated from the reaction of D5 with OH radicals, necessarily requiring the cleavage of Si-O bonds. Thus, the Si-O bonds may be cleaved from the OH radical attack, which may reduce the number of O atoms, leading to an increase of Si/O

at high PA. More detailed descriptions have been added in Lines 297-308 in the revised manuscript.

The calculation processes of the Si/O and Si/C ratios at each PA are shown as follows:

(1) The normalized peak intensity of each $C_xH_yO_zSi_n$ ion ($C_{x1}H_{y1}O_{z1}Si_{n1}$,

$C_{x2}H_{y2}O_{z2}Si_{n2}$ …$C_{xi}H_{yi}O_{zi}Si_{ni}$) is obtained from HR-ToF-AMS, which is named as $A_1$, $A_2$…$A_i$, respectively.

(2) The fraction of each $C_xH_yO_zSi_n$ ion ($F_1$, $F_2$, $F_3$…$F_i$) is calculated by Equation S1,

$$F_i = A_i \, / \, \text{MAX} \, (A_1, A_2, A_3 \ldots A_i) \tag{S1}$$

(3) The $C_xH_yO_zSi_n$ ions with $F_i \geq 0.01$ are used to calculate the ratio of Si/O and Si/C at each equivalent day by Equation S2 and S3, respectively.

$$n/z = \text{SUM} \, (F_i \times n_i/z_i) \, / \, \text{SUM} \, (F_i) \tag{S2}$$

$$n/x = \text{SUM} \, (F_i \times n_i/x_i) \, / \, \text{SUM} \, (F_i) \tag{S3}$$

When assigning molecular formulas to ion fragments, there are many candidates at one m/z.

The final molecular formula was determined based on the possible reaction process of cVMS

and OH radicals. Thus, the assignment of peaks may lead to the uncertainty of the results. The related descriptions have been stated in Line 288 in the revised manuscript and in Text S1 in the revised *Supplement*.

5) I thank the authors for their consideration of my previous comment regarding the back of the envelope calculations for the cVMS SOA loadings and for the addition of the discussion regarding the Milani and Pennington papers. However, I feel that my question regarding the choice of 8.5 days for urban sites was not sufficiently addressed. I agree with the authors that

8.5 days is within the lifetime of siloxanes, however, 8.5 days is longer than an airmass would remain in an urban area and thus I think these numbers are biased extremely high. The

Pennington et al data quoted in the paper (21 ng/m3) is much lower than the urban values reported in this paper and thus I am unsure of how the conclusion that the Pennington et al results are within the results reported here (lines 392-394) was reached. In my opinion, using

8.5 days is biasing the results extremely high and this choice needs to be both better justified and the implications of the choice (including biases) needs to be more transparently addressed (including the in the abstract).

**Re**: We agree that 8.5 days are longer for the residence time of an airmass in an urban area. It is reasonable that the SOA yields under high-$NO_x$ and seeded conditions are employed in the calculation of cVMS SOA concentrations at urban sites. The SOA formation from the reaction of cVMS with OH would always occur when the airmass is transported from urban aeras to low-$NO_x$ sites such as rural, forested, and polar regions. As shown in Figure 1 in the revised manuscript, the SOA yields under high-$NO_x$ conditions were generally smaller than those at similar OH exposures under low-$NO_x$ conditions. To simplify the estimation process, the SOA

yields at the maximum equivalent days (i.e., 8.5 days) here were used for the calculation of

SOA generated by cVMS from urban areas. It should be noted that these estimations may have large deviations, and they can still predict the SOA production from cVMS. The related descriptions have been stated in Lines 32-34, 368-372 and 389-392 in the revised manuscript.

The concentrations of D5 SOA in 14 urban sites have been estimated in our work. D5 SOA

concentrations ranged from 13.4 to 684 $ng/m^3$, which covered the D5 SOA concentration (21

$ng/m^3$) reported by Pennington et al (*Atmos. Chem. Phys.*, 2021, 21, 18247-18261). Therefore, it was stated that the model data (21 $ng/m^3$) of Pennington et al (*Atmos. Chem. Phys.*, 2021, 21,

18247-18261) was within the results reported here. The related descriptions have been modified in Lines 433-435 in the revised manuscript.

**Technical Comments**

Line 70-71: I thank the authors for their addition to the manuscript in response to my earlier comment. For the reader not familiar with the siloxane literature though, the addition could be difficult to follow. I encourage the authors to consider revising to something perhaps bigger picture about how $O_3$ and $NO_3$ loss pathways are low, but Cl needs to be evaluated more thoroughly due to potential spatial and temporal overlap.

**Re**: The loss of cVMS in the atmosphere is negligible through $O_3$ and $NO_3$ due to their small reaction rates (Atkinson, 1991). The global loss by the reaction with Cl atoms is less than 5 %

on account of low Cl concentrations, although it may be higher in some regions where cVMS

emissions and Cl sources overlap in both space and time (Alton and Browne, 2020). The related descriptions have been modified in Lines 72-76 in the revised manuscript.

Line 149: I believe the reference here should be to (Peng et al., 2018) for the high $NO_x$

conditions.

**Re**: We have checked carefully the reference in Line 156 in the revised manuscript, and it indeed refers to Peng et al (*Atmos. Chem. Phys.*, 2019, 19, 813-834).

Line 246-247: I think this sentence would benefit from more nuance. Not all of the D5

oxidation products are non-volatile. In fact, the early generation ones are volatile enough that they do not partition to aerosol in any significant way (e.g. Alton and Browne, 2020, 2022)

**Re**: The description has been modified into "In fact, most D5 oxidation products have been shown to be nearly non-volatile", and these two references above have been cited in Lines 256-

257 in the revised manuscript.

Lines 316-317: I did not notice this in my initial review, but on this reading it appears to me that this sentence is implying the formation of organic nitrates rather than inorganic nitrates.

How is the formation of some form of inorganic nitrate ruled out? There are no NH4+ peaks in

Fig. S9, but it is unclear if that is because they were not measured, or just not included in the figure. Are the ratios of $NO^+/NO_2^+$ different than an ammonium nitrate standard?

**Re**: According to previous studies, the ratio of $NO_2^+$ (m/z 46) to $NO^+$ (m/z 30) in the mass spectra detected by AMS can be used to distinguish organic or inorganic nitrates (Zhao et al.,

*Atmos. Chem. Phys.*, 2018, 18, 1611-1628; Fry et al., *Atmos. Chem. Phys.*, 2018, 18, 11663-

11682; Boyd et al., *Environ. Sci. Technol.* 2017, 51, 7831-7841; Ng et al., *Atmos. Chem. Phys.*,

2007, 7, 5159-5174). Organic nitrates are usually considered to have a $NO_2^+/NO^+$ of ~0.1-0.175, which is typically 2-3 times lower than that for $NH_4NO_3$ (0.31-0.85) (Zhao et al., *Atmos. Chem.*

*Phys.*, 2018, 18, 1611-1628; Fry et al., *Atmos. Chem. Phys.*, 2018, 18, 11663-11682; Fry et al.,

*Atmos. Chem. Phys.*, 2009, 9, 1431-1449; Boyd et al., *Environ. Sci. Technol.* 2017, 51, 7831-

7841; Fry et al., *Atmos. Chem. Phys.*, 2013, 13, 8585-8605). In our study, taking D3-D6 SOA

at PA of 6.9 d in high-$NO_x$ experiments as examples, $NO_2^+/NO^+$ ranged from 0.015 to 0.66, indicating that both inorganic and organic nitrates may be generated in high-$NO_x$ experiments.

The related description has been modified in Line 336 in the revised manuscript. There are no

$NH_4^+$ peaks in Figure. S9, because they were not measured.

Table S4: Are the mass concentrations based on SMPS + density or AMS. Granted, they should agree, but I think it should be specified. Given the results in Fig. S3 showing the variability within an experiment, some indication of variability in these mass concentrations should be included.

**Re**: The mass concentrations of D5 SOA in Table S4 were obtained on the basis of SMPS and effective aerosol density. In addition, the mass concentrations of cVMS SOA are the average data of SMPS in the last 10 minutes for each OH exposure. The related description has been specified in the title of Table S4 in the revised *Supplement*.

**References**

Alton, M. W. and Browne, E. C.: Atmospheric Chemistry of Volatile Methyl Siloxanes: Kinetics and Products of Oxidation by OH Radicals and Cl Atoms, Environ. Sci. Technol., 54, 5992–5999, https://doi.org/10.1021/acs.est.0c01368, 2020.

Alton, M. W. and Browne, E. C.: Atmospheric Degradation of Cyclic Volatile Methyl Siloxanes: Radical Chemistry and Oxidation Products, ACS Environ. Au, https://doi.org/10.1021/acsenvironau.1c00043, 2022.

Peng, Z., Palm, B. B., Day, D. A., Talukdar, R. K., Hu, W., Lambe, A. T., Brune, W. H., and Jimenez, J. L.: Model Evaluation of New Techniques for Maintaining High-NO Conditions in Oxidation Flow Reactors for the Study of OH-Initiated Atmospheric Chemistry, ACS Earth Space Chem., 2, 72–86, https://doi.org/10.1021/acsearthspacechem.7b00070, 2018.

Rücker, C. and Kümmerer, K.: Environmental Chemistry of Organosiloxanes, Chem. Rev., 115, 466–524, https://doi.org/10.1021/cr500319v, 2015.

**Re**: We thank the reviewer for providing these literatures. We have read them carefully and some of them have been cited and discussed properly in the revised manuscript.

Sincerely yours,

Chong Han, Professor

School of Metallurgy, Northeastern University

Shenyang 110819, China

E-mail: hanch@smm.neu.edu.cn

Shao-Meng Li, Chair Professor

College of Environmental Sciences and Engineering, Peking University

Beijing, China 100871

E-mail: shaomeng.li@pku.edu.cn

---

## Author Response (AR3)

Dear Professor Jason Surratt,

We welcome the opportunity to revise and clarify our manuscript for publication in

*Atmospheric Chemistry and Physics*. Below is a point-by-point response to the comments of reviewers.

**Anonymous referee #2**

Although the revision has been improved, I am not still fully convinced that the results can be applied to the environmentally relevant conditions. Charan et al. (2022) claimed that the yield of SOA depended strongly on OH concentration rather than OH exposure. Since the authors used a fixed residence time of 2 min in the OFR, the exposure reported was a directly proportional to OH concentration, which should be the governing factor for SOA formation.

The OH exposure (OH concentration multiplied by exposure duration) and photochemical age can be used interchangeably using an assumption of a constant OH concentration. Under an ideal condition, a low OH concentration for a long exposure time and a high OH concentration for a short exposure time can achieve the same degree of degradation of the parent compound.

However, that is not for the formation of SOA as Charan et al. (2022) asserted. Since OH

concentrations in the OFR were >1,000 times greater than the average OH concentration in air, the result of SOA formation yield should not be applied to the prediction in the real atmosphere.

The predicted SOA concentrations were overestimated when compared with other predictions in the literature. Thus, the current section of 4 Conclusions and implications should be properly revised.

**Re**: We agree that the SOA yield depends on the OH concentration. Charan et al. also claimed that the SOA yield varied in a small range (0-6%) at the OH concentration $\leq 5 \times 10^8$ molecules cm$^{-3}$ (Atmos. Chem. Phys., 2022, 22, 917-928). It was assumed that the SOA yields at environmentally relevant OH concentrations ($\sim 10^6$ molecules cm$^{-3}$) were similar to those at the lowest OH concentrations ($\sim 10^8$ molecules cm$^{-3}$) used here. Thus, the high-NO$_x$ SOA yields (D3: 0.038; D4: 0.001; D5: 0.011; D6: 0.000) under the seeded conditions at the OH

concentration of $6.83 \times 10^8$ molecules cm$^{-3}$ (0.63 equivalent days) were employed in the calculation of cVMS SOA concentrations at urban sites, while the low-NO$_x$ SOA yields (D3:

0.041; D4: 0.013; D5: 0.023; D6: 0.004) under the unseeded conditions at the OH concentration of $4.57 \times 10^8$ molecules cm$^{-3}$ (0.42 equivalent days) were used to estimate cVMS SOA at background and polar sites.

It was noted that the time-integrated consumption over the lifetime of a precursor compound in the atmosphere should be suitable for evaluating the SOA formation. As you stated, under an ideal condition, a low OH concentration for a long exposure time and a high OH concentration for a short exposure time can achieve the same degree of degradation of the precursor compound. Accordingly, when considering half-lives (6-30 days) of cVMS in the atmosphere, the $\Delta C_{cVMS}/C_{in-cVMS}$ values in low-NO$_x$ and unseeded experiments at 14.2 equivalent days (Table S7) are used to calculate cVMS SOA at background and polar sites. Due to a short residence time of airmass over urban areas, the $\Delta C_{cVMS}/C_{in-cVMS}$ values in high-NO$_x$ and seeded experiments at 0.63 equivalent days (Table S7) are employed to estimate cVMS SOA at urban sites.

The related descriptions have been given in Lines 371-377, 379-382 and 384-392 in the revised manuscript. The section of conclusions and implications has been properly modified on the basis of the updated data in Figure 4 in the revised manuscript, and in Tables S6-7 in the revised *Supplement*.

**Anonymous referee #3**

**Main Points**

1) I thank the authors for their additional text regarding the back of the envelope calculations on cVMS SOA concentrations. I think these types of calculations are valuable. However, I am still unconvinced by the arguments regarding the choice of 8.5 days. I agree that the SOA formation will occur as the airmass is transported and that for this type of calculation simplification is required. I disagree though that 8.5 days is the appropriate choice. As the airmass is transported it will undergo dilution in addition to chemical processing. For the equations used here, the aerosol is being calculated in terms of ng/m$^3$ and the dilution would not be taken into account when using 8.5 days and measured gas-phase concentrations near the source region. I think a more appropriate choice, particularly for the urban environment, would be to select a yield associated with the photochemical age appropriate for an urban airmass (~1

day or less). The impact of dilution on the results should be discussed.

**Re**: There is a short residence time of airmass over urban areas. The cVMS SOA yields (D3:

0.038; D4: 0.001; D5: 0.011; D6: 0.000) and the $\Delta C_{cVMS}/C_{in-cVMS}$ values in high-$NO_x$ and seeded experiments at 0.63 equivalent days have been employed in the calculation of cVMS

SOA concentrations at urban sites. The dilution of cVMS would occur during the transportation in the atmosphere, leading to an uncertainty of cVMS concentrations ($C_{cVMS}$) in the Equation

4. To simplify the estimation process, the effect of dilution on $C_{cVMS}$ would not be taken into account, and the reported $C_{cVMS}$ values were directly used. The related descriptions have been given in Lines 368-371, 375-377 and 387-389 in the revised manuscript. The section of conclusions and implications has been properly modified on the basis of the updated data in

Figure 4 in the revised manuscript, and in Tables S6-7 in the revised *Supplement*.

2) Si/O and Si/C ratio calculation. I thank the authors for the additional text clarifying the calculation of the Si/O and Si/C ratios. However, I think the text requires a bit more elaboration to make the procedure clearer. From the way the calculation is written, it appears that the number of Si or O in each ion is being used rather than the mass of Si and O. I think it is also more appropriate to calculate the mass of Si and the mass of O separately then calculate the

Si/O making the appropriate adjustments for atomic ratios (Aiken et al., 2007). It is also unclear to me why in equation S1 max is used in the denominator rather than sum if the fraction is being calculated. Additionally, uncertainty in this calculation goes beyond simply uncertainty in assigned peaks, but also includes fragmentation (see Aiken et al paper referenced above).

**Re**: The calculation methods of Si/O and Si/C ratios have been modified as follows:

(1) The peak intensity of $C_xH_yO_zSi_n$ ($C_{x1}H_{y1}O_{z1}Si_{n1}$, $C_{x2}H_{y2}O_{z2}Si_{n2}$ …$C_{xi}H_{yi}O_{zi}Si_{ni}$) is obtained from the HR-ToF-AMS, which is named as $A_1$, $A_2$…$A_i$, respectively.

(2) The fraction of each $C_xH_yO_zSi_n$ ion ($F_1$, $F_2$, $F_3$…$F_i$) is calculated by Equation S1,

$$F_i = A_i \,/\, \mathrm{SUM}(A_1, A_2, A_3…A_i) \tag{S1}$$

(3) The Si/O and Si/C ratio of $C_xH_yO_zSi_n$ ions at each equivalent day are calculated by Equation

S2-S6,

$$m_{Si} = \text{SUM}(F_i \times n_i \times M_{Si} / M_{C_{xi}H_{yi}O_{zi}Si_{ni}}) \tag{S2}$$

$$m_O = \text{SUM}(F_i \times z_i \times M_O / M_{C_{xi}H_{yi}O_{zi}Si_{ni}}) \tag{S3}$$

$$m_C = \text{SUM}(F_i \times x_i \times M_C / M_{C_{xi}H_{yi}O_{zi}Si_{ni}}) \tag{S4}$$

$$n/z = \frac{m_{Si}}{M_{Si}} \Big/ \frac{m_O}{M_O} \tag{S5}$$

$$n/x = \frac{m_{Si}}{M_{Si}} \Big/ \frac{m_C}{M_C} \tag{S6}$$

where M is the molar mass of one specific element (Si, O, C and H) or $C_xH_yO_zSi_n$ ions; $m$ is the total mass of Si, O or C in all $C_xH_yO_zSi_n$ ions. The related descriptions have been given in

Text S1 in the revised *Supplement*.

For the calculation results, there may be some uncertainties due to the assignments of peaks in the HR-ToF-AMS and the fragmentation processes of the AMS ionization. The revised

Si/O and Si/C mass ratios have been updated in Figure S8 and S11. The related descriptions have been revised in Lines 287-288 in the revised manuscript.

**Technical: Line numbers refer to track changes version.**

Line 305: I believe this should be Si-C, not S-C.

**Re**: The S-C has been modified into Si-C in Line 299 in the revised manuscript.

Line 256-257: This statement is incorrect. The Alton and Browne papers say the opposite – the products measured there are predicted to have high vapor pressures that wouldn't partition to aerosol under most ambient conditions (assuming absorptive partitioning). Wu and Johnston also show that a significant fraction of the products have a logC* >2 making them also volatile.

**Re**: The original statement "In fact, most D5 oxidation products have been shown to be nearly non-volatile" has been deleted in the revised manuscript.

Line 433-435: The Pennington et al results were for aerosol from multiple different siloxanes, not just D5. I also find the statement somewhat misleading. While the numbers reported here do encompass the Pennington et al number, the lower end numbers here that are comparable to the Pennington et al numbers are for locations with a much lower population than Los Angeles. This difference is important given that VMS to some extent at least scales with population.

**Re**: The results of Pennington et al. (*Atmos. Chem. Phys.*, 2021, 21, 18247-18261) have been compared with total cVMS SOA at urban sites, as shown in Lines 396-399 in the revised manuscript.

**References**

Aiken, A. C., DeCarlo, P. F., and Jimenez, J. L.: Elemental Analysis of Organic Species with Electron Ionization High-Resolution Mass Spectrometry, Anal. Chem., 79, 8350–8358, https://doi.org/10.1021/ac071150w, 2007.

**Re**: We thank the reviewer for providing this literature. It has been cited properly in the revised *Supplement*.

Sincerely yours,

Chong Han, Professor

School of Metallurgy, Northeastern University

Shenyang 110819, China

E-mail: hanch@smm.neu.edu.cn

Shao-Meng Li, Chair Professor

College of Environmental Sciences and Engineering, Peking University

Beijing, China 100871

E-mail: shaomeng.li@pku.edu.cn